# Cortical tau deposition follows patterns of entorhinal functional connectivity in aging

Jenna N Adams[1]*, Anne Maass[1,2], Theresa M Harrison[1], Suzanne L Baker[3], William J Jagust[1,3]

[1]Helen Wills Neuroscience Institute, UC Berkeley, Berkeley, United States; [2]German Center for Neurodegenerative Disease, Magdeburg, Germany; [3]Lawrence Berkeley National Laboratory, Berkeley, United States

**Abstract** Tau pathology first appears in the transentorhinal and anterolateral entorhinal cortex (alEC) in the aging brain. The transition to Alzheimer's disease (AD) is hypothesized to involve amyloid-β (Aβ) facilitated tau spread through neural connections. We contrasted functional connectivity (FC) of alEC and posteromedial EC (pmEC), subregions of EC that differ in functional specialization and cortical connectivity, with the hypothesis that alEC-connected cortex would show greater tau deposition than pmEC-connected cortex. We used resting state fMRI to measure FC, and PET to measure tau and Aβ in cognitively normal older adults. Tau preferentially deposited in alEC-connected cortex compared to pmEC-connected or non-connected cortex, and stronger connectivity was associated with increased tau deposition. FC-tau relationships were present regardless of Aβ, although strengthened with Aβ. These results provide an explanation for the anatomic specificity of neocortical tau deposition in the aging brain and reveal relationships between normal aging and the evolution of AD.
DOI: https://doi.org/10.7554/eLife.49132.001

*For correspondence:
jnadams@berkeley.edu

## Introduction

Alzheimer's disease (AD) is characterized by amyloid-β (Aβ) plaques and hyperphosphorylated forms of the tau protein as neurofibrillary tangles (NFTs) (*Braak and Braak, 1991*). Both of these aggregated proteins can be found in the brains of cognitively normal older adults (OA), suggesting that they reflect the 10–25 year incubation period for AD (*Jack et al., 2010*). Using positron emission tomography (PET) and radiotracers that target Aβ and tau, we can now investigate these pathologies in vivo (*Schöll et al., 2019*), and examine how the deposition of these proteins in the aging brain may lead to AD.

Neither human PET studies nor autopsy data have pointed to a precise single focus where Aβ deposition begins, rather suggesting that this pathology appears multifocally and soon encompasses the majority of association cortex (*Braak and Braak, 1991*; *Palmqvist et al., 2017*; *Thal et al., 2002*; *Whittington et al., 2018*). In contrast, neuropathological studies show that cortical tau deposition begins focally in the entorhinal cortex (EC), specifically in the transentorhinal cortex, i.e. the transition between lateral portions of the EC and perirhinal cortex (*Braak and Braak, 1985*; *Braak and Braak, 1991*; *Kaufman et al., 2018*). Strikingly, tau pathology is found in the EC of the majority of OA, including those without concurrent Aβ pathology (*Braak and Braak, 1997*; *Maass et al., 2017*). The mechanisms that cause tau to spread out of the EC and to other cortical regions are key to understanding, and ultimately preventing, the development of AD.

Postmortem studies mapping tau deposition have generated inferences about how tau spreads through the brain, beginning in EC, and then progressing in a stereotyped spatiotemporal pattern

**eLife digest** The changes in the brain that cause Alzheimer's disease begin up to 25 years before the first symptoms appear. During this long incubation period, two proteins accumulate in brain tissue: amyloid-β and tau. Amyloid-β forms clumps known as plaques, while tau forms structures called tangles. But whereas amyloid plaques accumulate evenly throughout the brain, this is not the case for tau. Instead tau accumulates first within a region called the entorhinal cortex, which is important for memory. Findings in animals suggest that tau then spreads out of the entorhinal cortex to other brain regions through neural connections.

The entorhinal cortex itself consists of two subregions, which each accumulate tau at different times. The anterolateral subregion (or alEC for short) develops tau first, followed by the posteromedial subregion (pmEC). These two subregions process different types of memory and so have connections to different areas of the brain. Does tau therefore spread to brain regions connected to the alEC before it spreads to regions connected to the pmEC?

To test this prediction, Adams et al. scanned the brains of healthy young adults to map their brain connectivity patterns. Young adults were chosen because the aging process itself can alter this connectivity. The brains of healthy older adults, aged 60 or more, were then scanned to measure amyloid-β and tau. None of the older adults had cognitive symptoms of Alzheimer's disease. Despite this, many showed deposits of amyloid-β and tau in their brains. As predicted, alEC-connected regions contained more tau than pmEC-connected regions. Indeed, the stronger the connection between a brain region and the alEC, the more tau that region contained.

These relationships occurred in older adults with and without amyloid-β in their brains. However, they were stronger in the individuals with amyloid-β. This adds to evidence suggesting that amyloid-β promotes the spread of tau. Future experiments should measure how tau spreads within an individual's network of connections over time. In the long run, researchers may even find that therapies that stop tau from spreading out of the alEC could help prevent Alzheimer's disease from taking hold.

DOI: https://doi.org/10.7554/eLife.49132.002

first to temporal and limbic regions and then widely throughout association cortex (*Braak and Braak, 1991*). Both cross-sectional, and more recently, longitudinal PET studies have largely supported these 'Braak Stages' (*Cho et al., 2016*; *Harrison et al., 2019*; *Maass et al., 2017*; *Schöll et al., 2016*; *Schwarz et al., 2016*). Cellular and molecular data indicate that tau can spread trans-synaptically through axonal projections, driven by neuronal activity, and inducing pathology in downstream neurons (*de Calignon et al., 2012*; *Pooler et al., 2013*; *Wu et al., 2016*; *Yamada et al., 2014*). Both PET and molecular data indicate that this phenomenon is at least partly accelerated by Aβ (*Hurtado et al., 2010*; *Khan et al., 2014*; *Pooler et al., 2015*; *Schöll et al., 2016*; *Vemuri et al., 2017*).

This stereotypical pattern of tau distribution in conjunction with the molecular mechanisms of tau spread strongly suggest that tau progresses through the brain along neural pathways. As cortical tau pathology most likely originates in the EC, the patterns of tau spread may be mapped by tracing the connectivity of EC in large-scale neural networks. While the major projection of EC is to the hippocampus, it is also reciprocally connected with limbic and association cortex (*Van Hoesen, 1982*; *Swanson and Köhler, 1986*; *Witter et al., 1989*). The human EC contains two anatomically and functionally distinct subregions, the anterolateral EC (alEC) and the posteromedial EC (pmEC), which are the human homologues of the lateral and medial entorhinal areas described in rodents (*Maass et al., 2015*; *Navarro Schröder et al., 2015*). The alEC is more strongly connected to anterior temporal regions including perirhinal cortex and is involved in processing object-related memory, while the pmEC is more strongly connected to posterior medial regions and is involved in processing spatial memory (*Kerr et al., 2007*; *Ranganath and Ritchey, 2012*; *Schultz et al., 2012*; *Reagh and Yassa, 2014*; *Berron et al., 2018*). The alEC is particularly vulnerable to the effects of aging (*Olsen et al., 2017*; *Reagh et al., 2018*) and preclinical AD (*Khan et al., 2014*), which has been proposed to be due to its early susceptibility to tau pathology. In contrast, the pmEC seems to be largely spared from early tau pathology (*Braak and Braak, 1985*) and age- or preclinical AD-

related vulnerability (*Khan et al., 2014*; *Olsen et al., 2017*; *Reagh et al., 2018*), and may not become affected until later in the disease process. This natural dissociation of connectivity and tau deposition allows us to test the hypothesis that functional connectivity (FC) patterns of the earliest tau deposition region, that is the alEC, is a better predictor of subsequent cortical tau deposition than the pmEC.

The goal of this study was to examine whether tau spreads out of the EC through neuronal pathways in humans by contrasting whether FC networks derived from the alEC and pmEC were differentially associated with patterns of cortical tau deposition in OA. We generated distinct FC networks using seeds in each EC subregion as well as the entire EC with resting state fMRI in healthy young adults (YA). We chose to model FC in YA because of concerns that the networks generated in OA may have been modified by tau pathology (*Schultz et al., 2017*), and thus the YA networks are more likely to reflect healthy adult network structure. These FC networks were then used to examine patterns of tau deposition in the brain, measured with PET in cognitively normal OA. Our main hypothesis was that because tau originates in transentorhinal or lateral portions of the EC, cortical regions functionally connected to alEC should demonstrate more tau pathology than those connected to pmEC or in non-connected cortical regions. Furthermore, we hypothesized that the amount of tau deposition in a region would be proportional to the strength of alEC connectivity to that region, and that relationships between FC and tau would be strengthened in the presence of Aβ.

## Results

### Participants

Fifty-five YA (aged 20–35) and 123 cognitively normal OA (aged 60+) from the Berkeley Aging Cohort Study (BACS) were included in the study. All YA participants underwent structural and resting state functional 3T MRI. All OA participants received tau-PET with [18]F-Flortaucipir (FTP), Aβ-PET with [11]C-Pittsburgh Compound-B (PiB), and a standard cognitive assessment. A subset of OA (n = 87) also completed the same 3T MRI protocol as the YA participants, and their resting state fMRI data was used for supplemental FC analyses. Demographic information for each sample is presented in *Table 1*.

**Table 1.** Demographic information for the young adult (YA) and older adult (OA) samples.

|  | YA FC (n = 55) | OA FC (n = 87) | OA PET (n = 123) | OA FC vs. OA PET | |
|---|---|---|---|---|---|
|  | *M (SD)* or *n* (%) |  |  | *T* or $X^2$ | *P* |
| Age (years)[*] | 24.9 (4.4) | 77.5 (6.1) | 76.5 (6.5) | 1.17 | 0.24 |
| Sex (female) | 26 (47.3%) | 55 (63.2%) | 73 (59.3%) | 1.84 | 0.17 |
| Education (years) | 16.2 (1.9)† | 16.7 (1.9) | 16.9 (1.9) | −0.69 | 0.49 |
| MMSE | 29.3 (1.1)‡ | 28.7 (1.3) | 28.7 (1.2) | −0.28 | 0.78 |
| Global PiB† | N/A | 1.16 (0.24) | 1.15 (0.23) | 0.57 | 0.57 |
| Aβ+§ | N/A | 39 (45.3%) | 49 (40.2%) | 3.26 | 0.07 |
| APOE  ε4+¶ | N/A | 26 (30.6%) | 31 (25.8%) | 3.44 | 0.06 |

YA, Young adult; OA, Older adult; FC, functional connectivity; MMSE, Mini Mental State Exam; Aβ+, Aβ-positive participants (Global PiB DVR > 1.065); APOE, alolipoprotein E; *Age at FC or age at tau; †6 YA missing education; ‡3 YA missing MMSE; §1 OA missing PiB (in both FC and PET samples); ¶2 OA missing APOE from FC sample, 3 from PET sample.

DOI: https://doi.org/10.7554/eLife.49132.003

The following source data is available for  Table 1:

**Source data 1.** Source data for the demographic information presented in *Table 1*.

DOI: https://doi.org/10.7554/eLife.49132.004

## FC patterns of different entorhinal seeds are distinct

We investigated the resting state FC of three different entorhinal seeds. To investigate the full extent of the EC, including the transentorhinal region, we used a structural entorhinal seed derived from the FreeSurfer segmentation of each participant's native space MRI (*Figure 1a*). To investigate EC subregions, we used template space alEC and pmEC seeds defined in a previous study (*Maass et al., 2015*) (*Figure 1b*). We performed seed-to-voxel FC analyses using the CONN Toolbox (*Whitfield-Gabrieli and Nieto-Castanon, 2012*). First-level models were performed with semi-partial correlations, focusing on unilateral seeds and within-hemisphere FC to more accurately approximate EC neural pathways (see Materials and methods). Second-level results were obtained with one-sample t-tests controlling for age and sex, and thresholded at both the voxel (p<0.001) and cluster level (p<0.05, FDR correction).

Group level patterns of FC derived from the YA sample are depicted in *Figure 1c–f*. The EC (*Figure 1c*) was functionally connected to other medial temporal lobe structures such as the hippocampus, amygdala, and temporal pole, and lateral temporal lobe structures such as the middle and inferior temporal gyrus. FC was also found with regions such as the angular gyrus, posterior cingulate, and medial frontal cortex.

Patterns of alEC FC (*Figure 1d*) were largely limited to anterior temporal regions, such as the medial temporal lobe and inferior temporal gyrus, but also found in angular gyrus. Patterns of pmEC FC (*Figure 1e*) included posterior medial regions such as the parahippocampal gyrus, posterior cingulate, and precuneus, and was also found in the medial orbitofrontal cortex. FC patterns for the alEC and pmEC seeds were distinct, showing minimal spatial overlap (*Figure 1f*).

We repeated these analyses using the data from the 87 OA with fMRI (*Figure 1—figure supplement 1*). While the patterns were generally similar, the alEC connectivity was somewhat expanded into typical regions of pmEC connectivity, and pmEC connectivity was reduced. The alEC and pmEC connectivity also spatially overlapped to a greater extent.

Importantly, all subsequent analyses shown relating FC to tau utilized the YA FC data. However, we conducted parallel analyses using the OA FC to confirm and extend our findings, which are briefly described at the end of each section and provided in full as supplemental information.

## Tau preferentially deposits within regions of entorhinal FC

To test the hypothesis that tau deposition paralleled patterns of entorhinal FC, we extracted and binarized each YA FC map, removing each seed region from their respective FC mask to derive non-EC FC masks. For the EC seed, we compared tau deposition within regions of FC to cortical gray matter regions that did not demonstrate significant FC ('outside cortical regions'). For the alEC and pmEC subregions, we compared tau deposition between each FC mask and to outside cortical regions not included within either the alEC or pmEC FC masks. We quantified tau deposition in OA as the proportion of suprathreshold FTP voxels (>1.4 SUVR) within a region, which has been demonstrated as a reliable marker of AD-related tau pathology (*Maass et al., 2017*) and is not influenced by region size. We explored effects of Aβ by classifying each OA participant as Aβ- or Aβ+ based upon their global PiB DVR.

To compare tau deposition between EC FC regions and outside cortical regions, we performed a repeated measures ANCOVA. We contrasted tau deposition by including region (EC FC vs. outside cortical regions) as a within subjects factor, Aβ status as a between subjects factor, and age and sex as covariates. We found a significant main effect of region (F(1)=119.30, p<0.001; *Figure 2a*). Post-hoc paired t-tests indicated higher tau deposition within regions of EC FC compared to outside cortical regions (t(121)=10.01, p<0.001). We further found a significant region by Aβ status interaction (F(1)=10.38, p=0.002), such that Aβ+ participants had a greater mean difference in tau deposition in regions of EC FC compared to outside cortical regions than did Aβ- participants (t(73.57) = 3.03, p=0.003).

We next tested the hypothesis that alEC FC would be a better predictor of tau deposition than pmEC FC or outside cortical regions. We performed a repeated measures ANCOVA with region (alEC FC vs. pmEC FC vs. outside cortical regions) as a within subjects factor, Aβ status as a between subjects factor, and age and sex as covariates. We again found a significant effect of region (F(1.65) = 43.88, p<0.001). Post-hoc paired t-tests indicated higher tau deposition within regions of alEC FC compared to pmEC FC (t(121)=3.42, p=0.001), as well as for both alEC FC (t(121)=6.97,

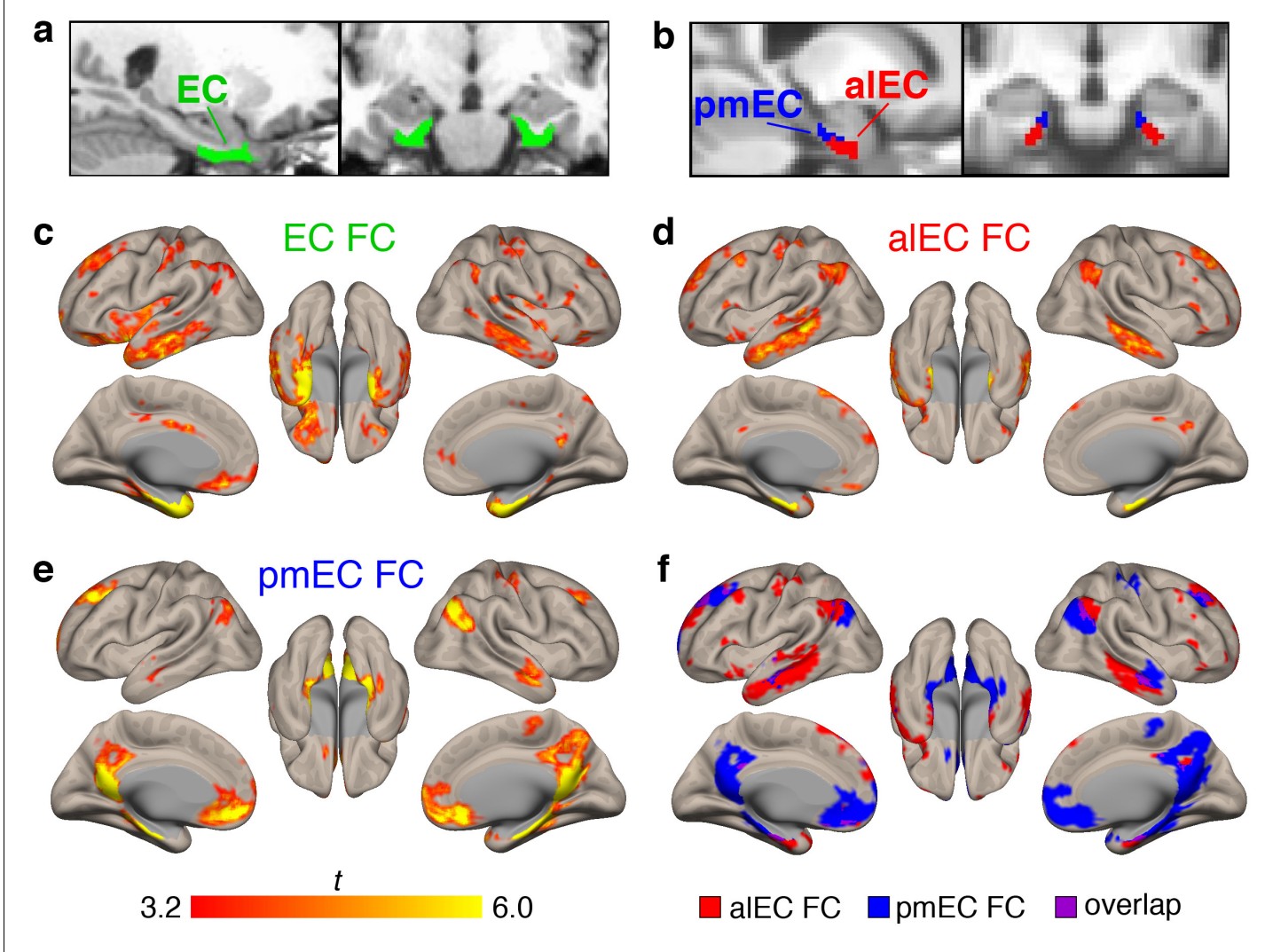

**Figure 1.** Functional connectivity (FC) of the different entorhinal seeds in healthy young adult (YA) participants. (**a**) The full entorhinal seed (EC, green), including transentorhinal, lateral, and medial regions, was derived from FreeSurfer segmentation of each participant's native space T1, and time series were extracted from native space fMRI data. (**b**) The anterolateral EC (alEC, red) and posteromedial EC (pmEC, blue) seeds were applied in template space, and time series were extracted before smoothing to preserve the spatial resolution of the seeds. (**c–e**) Seed-to-voxel FC analyses were performed for each seed with semi-partial correlations. Group level FC results were derived from one-sample t-tests controlling for age and sex, and thresholded at the voxel (p<0.001 uncorrected) and cluster level (p<0.05, FDR corrected). Results reflect t-statistics. (**c**) FC of the EC seed included medial temporal, lateral temporal, and limbic regions. (**d**) FC of the alEC seed included anterior temporal regions, such as medial and lateral temporal lobe. (**e**) FC of the pmEC seed included posterior medial regions, such as the parahippocampal gyrus and posterior cingulate. (**f**) Binary maps of alEC (red) and pmEC (blue) FC show little spatial overlap (purple) between the FC patterns. See *Figure 1—figure supplement 1* for parallel results using OA FC. See *Figure 1—figure supplement 2* for a visualization of gray matter voxels removed due to signal drop out, and alEC and pmEC seeds overlaid on the group-mean functional image.

DOI: https://doi.org/10.7554/eLife.49132.005

The following figure supplements are available for figure 1:

**Figure supplement 1.** Functional connectivity (FC) of the different entorhinal seeds in older adult (OA) participants.

DOI: https://doi.org/10.7554/eLife.49132.006

**Figure supplement 2.** Signal drop out demonstrated with group-mean functional images.

DOI: https://doi.org/10.7554/eLife.49132.007

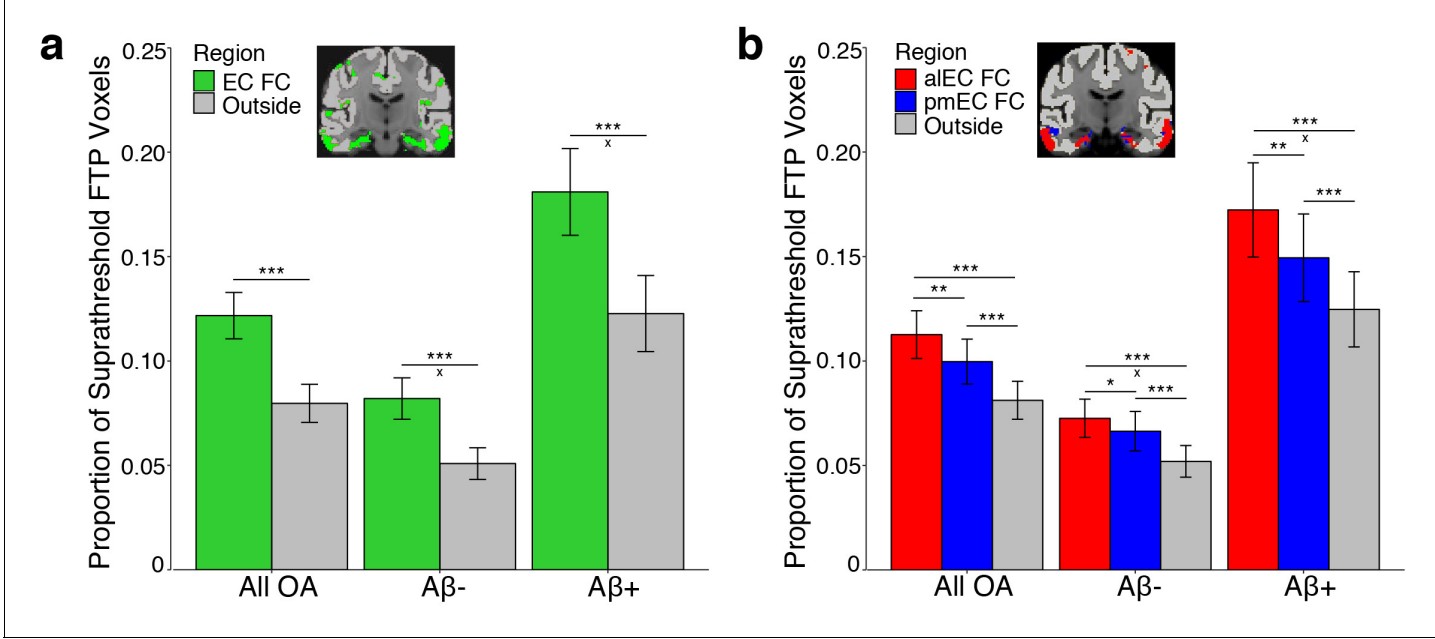

**Figure 2.** Tau preferentially deposits within regions of functional connectivity (FC) with entorhinal cortex. Tau deposition, defined as the proportion of suprathreshold FTP voxels (>1.4 SUVR), was measured in each FC mask and in cortical regions that did not demonstrate FC with the seeds ('Outside'). Tau deposition in each region was contrasted with repeated measures ANCOVAs and post-hoc t-tests. (a) Tau preferentially deposited within regions of EC FC (green) compared to outside cortical regions (gray). Region by Aβ status interactions were driven by an increased mean difference between the EC and outside cortical regions in the Aβ+ compared to the Aβ- group. (b) Tau preferentially deposited within regions of alEC FC (red) compared to both pmEC FC (blue) and outside cortical regions (gray), and in pmEC FC compared to outside cortical regions. Region by Aβ status interactions were driven by an increased mean difference between alEC FC and outside cortical regions in the Aβ+ compared to the Aβ- group. See *Figure 2—figure supplement 1* for parallel results using OA FC. ***p<0.001, **p<0.01, *p<0.05; 'x' indicates the drivers of significant Aβ status interactions (p<0.05); Error bars indicate the standard error of the mean.

DOI: https://doi.org/10.7554/eLife.49132.008

The following source data and figure supplements are available for figure 2:

**Source data 1.** Source data for the data visualized in *Figure 2*.
DOI: https://doi.org/10.7554/eLife.49132.011
**Figure supplement 1.** Tau deposition within regions of functional connectivity (FC) with entorhinal cortex derived from FC from the older adult (OA) participants.
DOI: https://doi.org/10.7554/eLife.49132.009
**Figure supplement 1—source data 1.** Source data for the data visualized in *Figure 2—figure supplement 1*.
DOI: https://doi.org/10.7554/eLife.49132.010

p<0.001) and pmEC FC (t(121)=6.66, p<0.001) compared to outside cortical regions. We also found a region by Aβ status interaction (F(1.65) = 6.09, p=0.005), which was driven by Aβ+ participants having a greater mean difference in tau deposition in regions of alEC FC compared to outside cortical regions than did Aβ- participants (t(59.53) = 2.64, p=0.01), while this difference in alEC FC compared to pmEC FC regions was trending (t(56.64) = 1.90, p=0.06).

We repeated these analyses using the OA FC masks and results were consistent with the YA results, except the region by Aβ status interaction in the subregion analysis was reduced to a trend (p=0.07) (*Figure 2—figure supplement 1*; *Supplementary file 1*).

## FC-specific tau deposition increases with Aβ and EC tau

Based on these results, both entorhinal FC and the presence of Aβ are related to the cortical distribution of tau. We next sought to further probe how the amount of cortical Aβ and EC tau influenced tau deposition at the FC targets of EC. We constructed 'FC specific' measures of tau deposition defined as the mean difference per participant between suprathreshold FTP voxels in 1) each of the

three FC masks compared to outside cortical regions and 2) alEC FC compared to pmEC FC regions.

We first assessed the relationship between FC specific tau and continuous levels of Aβ across the cortex with a measure of global PiB DVR. As global PiB DVR increased, FC specific tau deposition increased within regions of EC FC (r = 0.37, p<0.001), alEC FC (r = 0.47, p<0.001), and pmEC FC (r = 0.20, p=0.03), as well as within alEC FC compared to pmEC FC (r = 0.41, p<0.001). We repeated this analysis within only Aβ+ participants to ensure that associations were not driven by a floor effect of global PiB DVR. All associations remained significant (p's < 0.05) except for pmEC FC-specific tau deposition (p=0.26; *Supplementary file 2*). This finding indicated a strong relationship between continuous Aβ levels and proportionally greater tau deposition within EC and alEC FC targets that occurred across the Aβ+ spectrum.

Although Aβ is diffusely distributed in cortex, we next examined whether its location in the cortical connectivity targets of EC was important in driving tau spread. We quantified the mean PiB DVR within each FC mask for each participant. Across participants, PiB DVR was significantly higher within regions of pmEC FC compared to alEC FC (paired samples t-test, t(121)=26.61, p<0.001) and to EC FC (t(121)=18.92, p<0.001). However, associations between FC specific tau deposition and PiB DVR within the FC masks were of similar strength to that of global PiB DVR (*Supplementary file 2*), and therefore did not offer more precise information about tau deposition in these FC regions. PiB DVR within the FC masks was highly correlated with global PiB DVR (r's > 0.99), which may explain the similarity of the findings. Again, all results remained significant when analyzing only the Aβ+ participants, except for pmEC FC specific tau deposition (*Supplementary file 2*).

Finally, we sought to determine whether higher levels of EC tau were associated with proportionally greater tau deposition within its FC targets, with the hypothesis that more tau would be available to spread from the EC to downstream regions. We therefore examined the relationship between mean EC FTP SUVR and FC-specific tau deposition across subjects. As EC FTP SUVR increased, FC-specific tau deposition increased within regions of EC FC (r = 0.62, p<0.001), alEC FC (r = 0.46, p<0.001), and pmEC FC (r = 0.30, p=0.001), as well as within alEC FC compared to pmEC FC (r = 0.33, p<0.001). We repeated this analysis controlling for global FTP (mean SUVR across the cortex) to try to further isolate the effects of EC tau. All correlations remained significant (p's < 0.02) except for pmEC FC-specific tau deposition (p=0.59; *Supplementary file 2*), indicating that EC tau has a stronger relationship with tau deposition within regions of EC and alEC FC than within pmEC FC.

We repeated the Aβ and EC tau analyses using the OA FC masks, and the overall pattern of results was similar (*Supplementary file 2*).

## Stronger FC is associated with higher levels of tau deposition

We next investigated whether stronger average FC between an entorhinal seed and a region was associated with higher levels of tau deposition in that region. We subdivided each seed's FC mask into regions of low, medium, and high FC based on the YA group-average FC strength (beta value) in each voxel using one-dimensional k-means clustering (see Materials and methods). Results of this clustering are depicted in *Figure 3a–c*. We then calculated the proportion of suprathreshold FTP voxels within each FC strength region for each seed's FC mask, as this tau measure was not influenced by the different FC strength region sizes.

To test whether regions of stronger FC had a higher level of tau deposition, we performed a repeated measures ANCOVA within each seed's FC mask separately. We contrasted tau deposition between FC strength regions (low vs. medium vs. high FC) as a within subjects factor, included Aβ status as a between subjects factor, and age and sex as covariates. We found significant main effects of FC strength for the EC (F(1.14) = 73.29, p<0.001; *Figure 3d*), alEC (F(1.05) = 29.99, p<0.001; *Figure 3e*), and pmEC (F(1.22) = 33.55, p<0.001; *Figure 3f*) seeds. For the EC and alEC, stronger FC was associated with an increase in tau deposition in a stepwise fashion, with low < medium < high FC (post-hoc paired t-tests, p's < 0.001). However, we found the inverse association for the pmEC, where stronger FC was associated with a decrease in tau deposition (p's < 0.05).

We additionally found a FC strength by Aβ status interaction for the EC (F(1.14) = 15.16, p<0.001), alEC (F(1.05) = 8.43, p=0.004), and pmEC (F(1.22) = 4.46, p=0.03) seeds. For the EC, the difference in tau deposition between all FC strength regions was greater in the Aβ+ compared to Aβ- participants (independent samples t-tests, all p's < 0.01). For the alEC, this interaction was

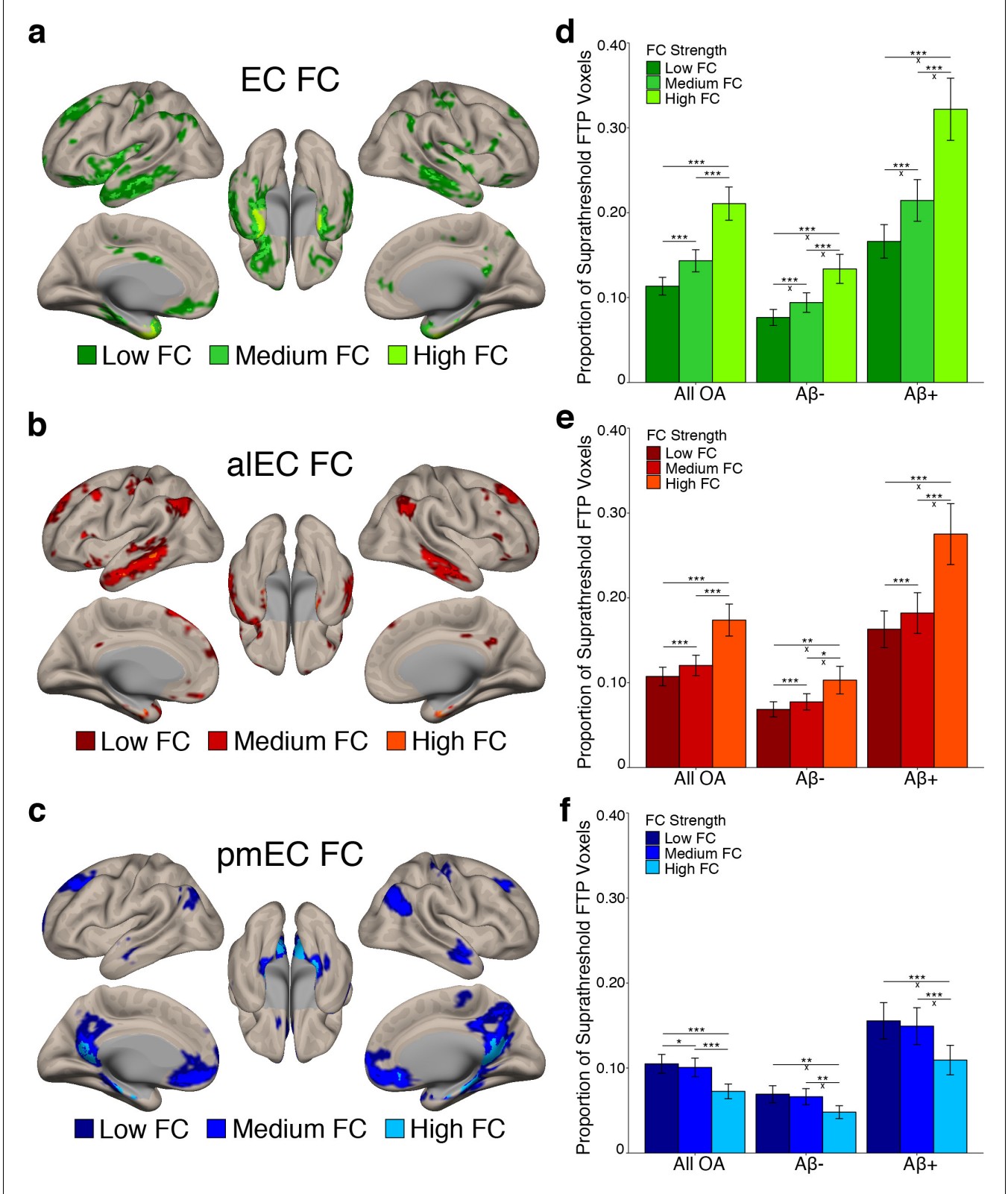

**Figure 3.** Stronger FC with a region is associated with higher levels of tau deposition in that region. (a-c) To examine associations between tau and FC strength, group level YA FC masks for each EC seed were clustered into regions of low, medium, and high FC based upon the mean YA FC strength (beta value) of each voxel. (d-f) The proportion of suprathreshold FTP voxels (>1.4 SUVR) was calculated for each FC strength region ('tau deposition'). Repeated measures ANCOVAs and post-hoc t-tests were performed to contrast tau deposition between each FC strength region. (d) EC FC strength

*Figure 3 continued on next page*

*Figure 3 continued*

was significantly related to tau deposition in a stepwise pattern, with tau increasing in low < medium < high FC regions. Aβ status interactions were observed between all comparisons. (**e**) Increasing alEC FC strength was also related to higher tau deposition in a stepwise fashion, and Aβ interactions were driven by the difference between high-medium and high-low FC regions. (**f**) Increasing pmEC FC strength was associated with decreases in tau deposition, and Aβ interactions were driven by the difference between high-medium and high-low FC regions. See *Figure 3—figure supplement 1* for parallel results using OA FC. ***p<0.001, **p<0.01, *p<0.05; 'x' indicates the drivers of significant Aβ interactions (p<0.05). Error bars indicate the standard error of the mean.

DOI: https://doi.org/10.7554/eLife.49132.012

The following source data and figure supplements are available for figure 3:

**Source data 1.** Source data for the data visualized in *Figure 3*.

DOI: https://doi.org/10.7554/eLife.49132.015

**Figure supplement 1.** Stronger older adult (OA) functional connectivity (FC) to a region is associated with higher levels of tau deposition to that region.

DOI: https://doi.org/10.7554/eLife.49132.013

**Figure supplement 1—source data 1.** Source data for the data visualized in *Figure 3—figure supplement 1*.

DOI: https://doi.org/10.7554/eLife.49132.014

mainly driven by a greater tau deposition difference in the Aβ+ compared to Aβ- participants in high FC compared to medium and low FC (p's < 0.05), while the medium to low FC comparison was trending (p=0.051). These interactions indicate that Aβ is specifically associated with more tau deposition within regions of stronger EC and alEC connectivity rather than regions of lower connectivity. However, for the pmEC, the interaction showed that while both the Aβ- and Aβ+ groups had the least amount of tau deposition in the high FC regions, the difference between low-high and medium-high was larger for the Aβ+ than the Aβ- subjects (p's < 0.05), while there was no significant difference in the medium-low FC comparison across groups (p>0.05). This interaction indicates that Aβ is associated with more tau deposition within regions of low and medium pmEC connectivity rather than regions of higher pmEC connectivity.

These analyses were repeated using the OA FC masks and provided largely consistent results, except that the interaction between pmEC FC strength and Aβ status was reduced to a trend (p=0.09; *Figure 3—figure supplement 1*; *Supplementary file 3*).

## Associations between EC tau and cortical tau increase with FC strength

We found that stronger FC with EC is related to higher cortical tau deposition, and that the amount of EC tau is related to proportionally greater tau deposition within its FC targets, but it was not clear whether the correlation between EC tau and cortical tau was directly related to the FC strength. To test this, we began by performing a voxelwise regression across all OA participants using EC FTP SUVR as the independent variable and cortical FTP SUVR within each voxel as the dependent measure, controlling for age and sex (see Materials and methods). This analysis produced a group-level EC-cortical tau association map that shows voxels in the brain where tau is significantly positively correlated with the amount of tau in the EC across subjects (*Figure 4a*). The strongest associations were in the medial temporal lobe, middle and inferior temporal gyrus, and posterior parietal lobe, while other significant regions included the retrosplenial cortex, precuneus, and frontal cortex.

To determine whether stronger FC between the EC and a voxel was correlated with a stronger association between EC tau and that same voxel's tau, we performed a voxelwise correlation between the group-average YA FC strength (beta value) of each voxel, and the group-average EC-cortical tau association strength (beta value) at each voxel. We found a significant positive correlation between EC FC strength and EC-cortical tau association strength (r = 0.44, rs = 0.29, p's < 0.001; *Figure 4b*), indicating that stronger FC from the EC to a voxel is related to more EC-associated tau in that voxel. We similarly found a positive correlation between alEC FC strength and EC-cortical tau association strength (r = 0.29, rs = 0.15, p's < 0.001; *Figure 4c*). While pmEC FC strength was also correlated with EC-cortical tau association strength, the relationship was very weak (r = 0.08, rs = 0.04, p's < 0.001; *Figure 4d*). These results replicated when using the OA group FC strengths (*Figure 4—figure supplement 1*).

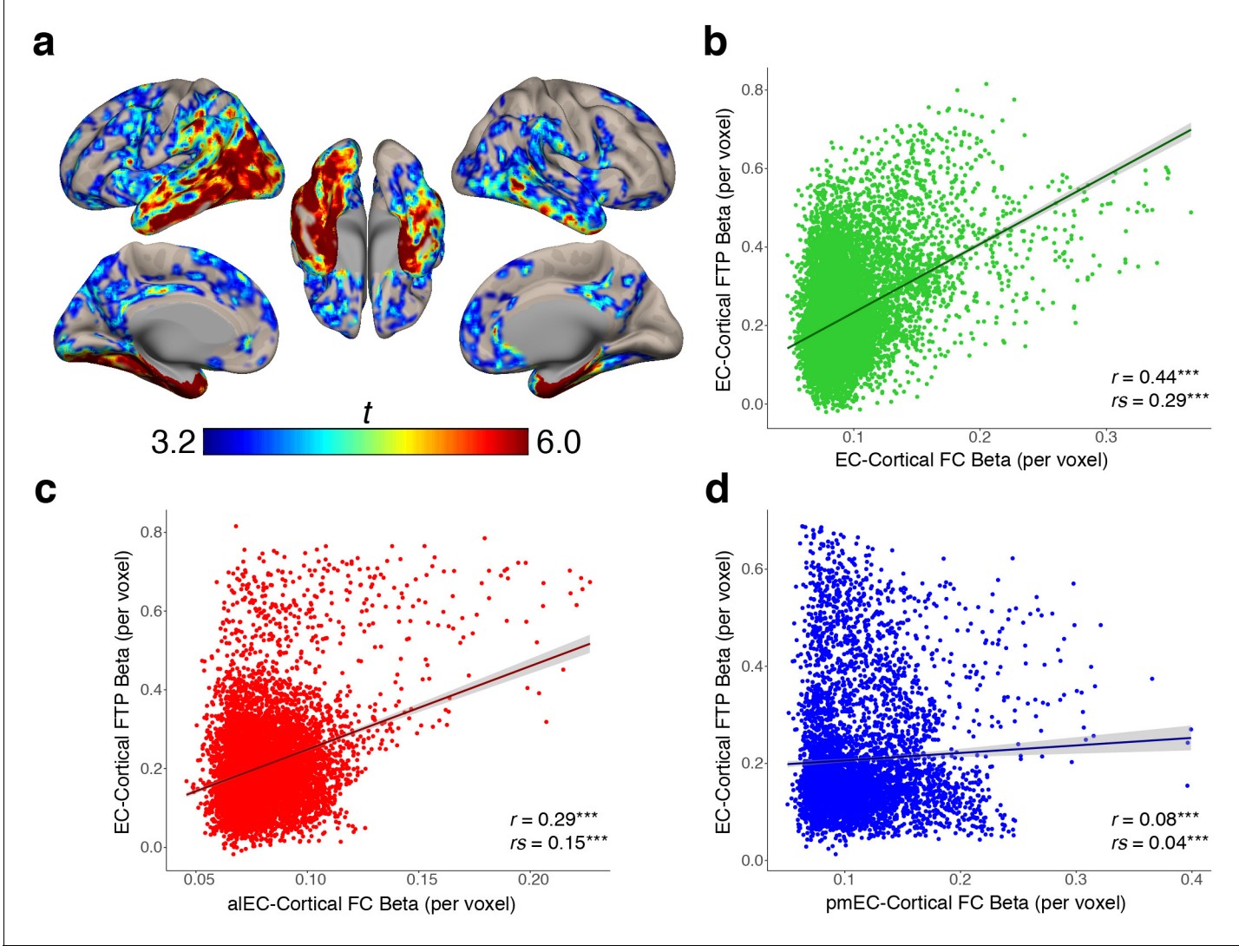

**Figure 4.** EC-cortical tau associations and relationships with FC strength. (a) Voxelwise regressions were performed across OA participants between the mean EC FTP SUVR predicting voxelwise FTP. One-sample t-tests show a group level map depicting voxels that had strong positive associations between their FTP and the amount of FTP in the EC. (b–d) YA average FC strength per voxel (beta) was correlated with each voxel's EC-cortical tau association strength (beta). FC of the EC (b) and alEC (c) seeds demonstrated positive associations, where stronger FC to a voxel was correlated with a stronger association between EC FTP and that voxel's FTP. The pmEC (d) showed a weak association. See *Figure 4—figure supplement 1* for parallel results using OA FC. ***p<0.001.

DOI: https://doi.org/10.7554/eLife.49132.016

The following figure supplement is available for figure 4:

**Figure supplement 1.** EC-cortical tau associations and relationships with functional connectivity (FC) strength from older adult (OA) participants.

DOI: https://doi.org/10.7554/eLife.49132.017

## Discussion

In this study, we demonstrate that patterns of EC FC predict the spatial topography and level of cortical tau deposition in a sample of cognitively normal OA, where regions with high entorhinal connectivity have more tau. Moreover, connectivity of the alEC subregion more strongly predicted tau in connected cortical regions than pmEC connectivity. These associations were not dependent on the presence of Aβ, however, higher Aβ increased the strength of FC-tau relationships. Additionally, the relationship between FC and cortical tau was closely associated to the amount of tau within EC, suggesting that as tau in EC increases, it begins to spread to connected cortex. Together, these

results support a model in which early tau in transentorhinal or lateral portions of the EC spreads to downstream cortical regions through functional connections, accelerated by the presence of Aβ.

Using resting state fMRI from YA participants, we obtained healthy patterns of FC from three different entorhinal seeds. The full EC seed, which included the transentorhinal region, was functionally connected to regions that are known to receive structural projections from the EC, including temporal and limbic regions (*Van Hoesen et al., 1975*; *Witter et al., 1989*). This pattern agreed with previous work showing entorhinal FC to the default mode network (*Huijbers et al., 2014*). The alEC and pmEC subregions demonstrated distinct FC patterns to anterior temporal and posterior medial regions, respectively, which parallel their unique structural connectivity (*Witter et al., 1989*) and functional involvement in the processing of object versus spatial information (*Ranganath and Ritchey, 2012*; *Reagh and Yassa, 2014*). These FC patterns were largely consistent with a previous study investigating cortical FC of the alEC and pmEC (*Navarro Schröder et al., 2015*), and were similar to FC patterns of their closely associated regions: the perirhinal cortex and parahippocampal gyrus (*Kahn et al., 2008*; *Libby et al., 2012*). The concordance of our FC networks with anatomical literature and previous FC findings increased our confidence in relating these FC patterns to tau deposition, which was the primary goal of this study.

Both the spatial extent and strength of EC and alEC FC was associated with the level of tau deposition in cognitively normal OA. The initial site of cortical tau deposition appears to be transentorhinal cortex and lateral EC (*Braak and Braak, 1985*; *Kaufman et al., 2018*), which is consistent with other reports suggesting that alEC is particularly vulnerable to effects of age and preclinical AD (*Berron et al., 2018*; *Khan et al., 2014*; *Reagh et al., 2018*). Our results indicate that alEC may serve as an epicenter of tau pathology, initiating propagation to distant areas of cortex in a topographically specific manner reflecting its connectivity. The idea that tau spreads trans-synaptically via large scale neural networks also has support from other laboratories which have indicated that nodes with higher FC show more tau or higher covariance of tau deposition (*Cope et al., 2018*; *Franzmeier et al., 2019*). Evidence for the trans-synaptic spread of tau in humans has also been demonstrated using diffusion tensor imaging (*Jacobs et al., 2018*). Our findings expand upon this previous literature by indicating an anatomic specificity to the earliest stages of cortical tau deposition, and provide an explanation for why tau is preferentially deposited in the anterior temporal network in aging and AD (*Maass et al., 2019*).

Patterns of pmEC FC did not predict the location or amount of tau deposition as well as the EC or alEC, and its FC strength demonstrated an inverse relationship with levels of tau deposition. However, tau pathology does occur in the posterior medial network which are targets of pmEC connectivity, such as posteromedial cortex, although this is likely at a later stage than in alEC connected regions (*Braak and Braak, 1991*; *Cho et al., 2016*; *Harrison et al., 2019*; *Maass et al., 2017*; *Maass et al., 2019*; *Vemuri et al., 2017*). Because the alEC has reciprocal axonal projections to cortex, while pmEC mostly lacks these efferent projections (*Witter et al., 1989*), tau may indirectly spread from the alEC to posteromedial regions through multisynaptic projections. This is supported by a recent PET study in humans that suggests tau spreads from the hippocampus to the posterior parietal cortex via the cingulum bundle (*Jacobs et al., 2018*). These data suggest that tau deposition in the posterior medial system is a later stage, dependent on further downstream spread to the hippocampus. This is consistent with longitudinal inferences drawn from cross-sectional autopsy-based Braak staging (*Braak and Braak, 1991*).

The association between entorhinal FC and tau deposition did not require the presence of Aβ, as we observed associations within our Aβ- sample. This finding is consistent with another recent human neuroimaging study that observed relationships between whole-brain FC and tau covariance strength in OA without Aβ (*Franzmeier et al., 2019*), as well as animal data indicating that tau spreads trans-synaptically without Aβ (*de Calignon et al., 2012*; *Wu et al., 2016*). The finding that tau may spread out of the entorhinal cortex to connected cortical regions in the absence of Aβ is important to better understand the results of clinical treatments of AD. Recent Aβ-reducing therapies have failed to prevent AD from worsening (*Egan et al., 2018*; *Salloway et al., 2014*). This may be because removing Aβ does not prevent the spread of tau, which is more strongly related to neurodegeneration and cognitive impairment (*Jagust, 2018*).

However, while Aβ may not be necessary for tau to spread out of the EC, it may still accelerate this spread, as the tau-FC associations we observed were stronger in Aβ+ participants and increased continuously with Aβ levels. While this finding is consistent with other reports in the animal literature

(*Hurtado et al., 2010*; *Pooler et al., 2015*), stronger tau-FC relationships with increasing Aβ have not been previously demonstrated in humans using neuroimaging. Because local interactions between Aβ and tau in the EC are unlikely in a cognitively normal OA sample due to their separate spatial topographies (*Lockhart et al., 2017*; *Sepulcre et al., 2016*), we investigated whether the amount of Aβ at the cortical targets of the EC explained FC-related tau deposition. However, Aβ within the FC targets was not a better predictor of FC-related tau deposition than global levels of Aβ, although these two Aβ measures were highly correlated. It is possible that tau-Aβ interactions might be better explained by soluble oligomeric forms of Aβ not measured by PET, or by other unknown molecular interactions beyond the scope of this study.

We also found a strong relationship between the amount of EC tau and the association between FC and cortical tau. Increased tau deposition within the EC was associated with proportionally greater tau deposition within its cortical FC targets, particularly within alEC FC. Additionally, higher EC and alEC FC strength with a cortical target was correlated with a stronger association between EC tau and tau in that cortical target, indicating that the association between EC tau and cortical tau was related to the FC strength between them. Together, these findings suggest that as tau accumulates and increases within EC, it begins to spread to functionally connected cortex in a pattern consistent with its connectivity strength.

The alEC is particularly vulnerable to the effects of aging, largely reflecting its early role in the tau pathological cascade. In OA, structural and functional alterations in the alEC seem to be related to both age and AD-related pathology, while the pmEC does not show these same changes (*Berron et al., 2018*; *Olsen et al., 2017*; *Reagh et al., 2018*). As the alEC is one of the most interconnected regions of the brain (*Bota et al., 2015*; *Swanson and Köhler, 1986*), alEC tau pathology has the potential to spread to and disrupt widespread cortical regions. Therefore, characterizing the pattern of alEC connectivity and its relationship to tau deposition is a critical factor in understanding and potentially predicting the trajectory of tau spread. While the majority of OA have tau deposition within the EC (*Braak and Braak, 1997*; *Maass et al., 2017*; *Schöll et al., 2016*), the spread of tau to alEC-connected cortex may be one of the first signs of AD or a related neurodegenerative disease such as primary age related tauopathy (PART). Our finding that Aβ is associated with more tau deposition in alEC-connected cortex indicates that the earliest stage of AD, characterized by Aβ-facilitated tau spread, requires a background of normal aging, which partly explains the poorly understood relationship between normal aging and AD.

Because tau and Aβ both affect FC (*Schultz et al., 2017*), we chose to use YA FC to establish normal connectivity patterns and to avoid any tau-induced changes that may be late-life alterations that have little impact on long-term patterns of tau spread. It is important to note, however, that the results relating the OA FC data to tau deposition were highly similar. To our knowledge, patterns of alEC and pmEC FC have not previously been described in OA. It is interesting to note that alEC connectivity in OA expanded to encompass regions of posteromedial and prefrontal cortex that are normally associated with the posterior medial scene processing system, while pmEC FC was reduced in scope. This could be due to age-related dedifferentiation of the networks (*Goh, 2011*; *Maass et al., 2019*), or to atrophy causing more overlap between the seeds and thus increased similarity of the networks. Future examination of age and pathology related changes in these networks using more precise native-space subregions would be useful for better understanding cognitive aging.

There are several limitations of our study. First, we acknowledge that measurements of FC based upon fMRI data are indirect measures of neural activity. However, the BOLD signal is reliably associated with neural activity (*Fox and Raichle, 2007*), and resting state fMRI has been successfully used to measure FC in previous related studies (*Cope et al., 2018*; *Franzmeier et al., 2019*). We were unable to distinguish the directionality of FC, and thus our results may also represent activity directed to our seed regions rather than solely from projections. This lack of directionality also prevents us from determining whether the patterns of tau deposition we observed reflect anterograde or retrograde spread of tau out of the EC, which has important implications for tracking the progression of tau spread. Future studies should aim to differentiate these processes. Additionally, because FC measures may also reflect multisynaptic connections (*Fox and Raichle, 2007*), and tau can progress across multisynaptic connections (*Wu et al., 2016*), our results may include the spread of tau from other downstream regions.

Further, we note that the resolution of our 3T fMRI data was relatively low compared to previous high-resolution fMRI studies defining functional EC subregions and their connectivity (*Maass et al.,*

2015; Navarro Schröder et al., 2015). While we found anatomically distinct connectivity patterns of the entorhinal subregions resembling previous results (Navarro Schröder et al., 2015), there was some minimal overlap between the networks. Although the patterns of our networks still explained vulnerability to tau deposition, the use of higher resolution fMRI may reveal more precise networks and therefore even better predictions for tau spread.

Due to the resolution of PET, we were unable to distinguish tau deposition in the EC subregions to confirm that alEC was affected sooner than pmEC; however, neuropathological data and the vulnerability of alEC compared to pmEC strongly suggests this occurrence (Braak and Braak, 1991; Khan et al., 2014; Reagh et al., 2018). Additionally, the [18]F-Flortaucipir tracer exhibits known off-target binding to the choroid plexus that prevents accurate measurements of tau deposition within the hippocampus (Baker et al., 2017). Thus, we were not able investigate specific associations between hippocampal tau and entorhinal FC.

Finally, the cross-sectional nature of these associations limits our interpretation of causality. While it is possible that tau deposition increases FC (rather than the reverse), this seems unlikely because of our use of FC data from YA, who do not have appreciable tau deposition. In addition, considerable evidence suggests that tau reduces neuronal activity and FC strength (Busche et al., 2019; Schultz et al., 2017). Thus, our data appear to support the first stage of a bidirectional relationship between tau and FC, where stronger FC initially guides tau spread, and tau deposition later reduces FC.

Our findings provide an explanation for the topographically specific patterns of neuronal vulnerability to tau deposition, suggesting this pattern reflects tau spread from the transentorhinal or lateral EC to functionally connected cortex. The observation that this process is not dependent on, although facilitated by, Aβ suggests that AD may develop from processes of normal aging. Efforts to develop effective treatments for AD are moving towards earlier and earlier detection, and away from Aβ-lowering therapies which have failed in clinical trials. The ability to detect the very earliest spread of tau may be crucial in selecting individuals for tau-directed therapies before symptoms of cognitive decline appear. In order to do this, the biology of how and where tau spreads needs to be better understood.

## Materials and methods

### Participants

Fifty-five young adults (YA) ages 20–35 were included for analysis. YA participants received structural and functional 3T MRI. YA were recruited for the study through flyers posted on the UC Berkeley campus and by word of mouth. Inclusion criteria included no major health problems, no current or recent history of psychiatric illness, no history of neurological disorders or traumatic brain injury, no substance abuse problems, no depression or antipsychotic medications, and fluency in English.

123 cognitively normal OA from the Berkeley Aging Cohort Study (BACS) were included for analysis. All OA participants received tau-PET imaging with [18]F-Flortaucipir (FTP) and Aβ-PET with [11]C-Pittsburgh Compound B (PiB), 1.5T structural MRI, and neuropsychological testing as part of normal BACS protocol. A subset of 87 OA also received structural and functional 3T MRI. Eligibility requirements for the BACS participants included age ≥60 years, cognitively normal status (Mini-Mental State Examination score ≥25 and normal neuropsychological examination, defined as within 1.5 SDs of age, education, and sex adjusted norms); no serious neurological, psychiatric, or medial illness; no major contraindications found on MRI or PET; and independent living in the community.

This study was approved by the Institutional Review Boards of the University of California, Berkeley, and the Lawrence Berkeley National Laboratory. All participants provided written informed consent.

### 3T functional and structural MRI

#### 3T MRI acquisition

All YA participants and a subset of OA participants received structural and functional MRI at the Henry H. Wheeler Jr. Brain Imaging Center. Data was acquired on a 3T TIM/Trio scanner (Siemens Medical System, software version B17A) using a 32-channel head coil. A whole brain high-resolution T1-weighted volumetric magnetization prepared rapid gradient echo image (MPRAGE) was first

acquired (voxel size = 1 mm isotropic, TR = 2300 ms, TE = 2.98 ms, matrix = 256×240 x 160, FOV = 256×240 x 160 mm$^3$, sagittal plane, 160 slices, 5 min acquisition time).

A resting state functional MRI was then collected using T2*-weighted echo planar imaging (EPI, voxel size = 2.6 mm isotropic, TR = 1067 ms, TE = 31.2 ms, FA = 45, matrix = 80×80, FOV = 210 mm, sagittal plane, 300 volumes, anterior to posterior phase encoding, ascending acquisition, 5 min acquisition time). A multiband acceleration factor of 4 was used to obtain whole brain coverage at high spatial resolution by acquiring four slices at the same time (*Feinberg and Setsompop, 2013*; *Todd et al., 2016*). During acquisition, participants were instructed to remain awake with their eyes open and focused on the screen, which displayed a white asterisk on a black background.

## Structural MRI processing

Structural 3T T1 MRIs were processed using Statistical Parametric Mapping 12 (SPM; www.fil.ion.ucl.ac.uk/sp). The T1 image was first segmented into gray, white, and CSF compartments in native space. DARTEL-imported tissue segments were used to create a study specific template to facilitate warping to MNI space, performed separately for the YA and OA participants. Native space T1 images and tissue segments were then warped to MNI space at 2 mm isotropic resolution using the group template created in the previous step. Additionally, a gray matter mask was created by averaging the MNI space gray matter images for the YA and OA groups, separately, and thresholded at >25% probability of gray matter voxels. Finally, native space T1 images were segmented with FreeSurfer v.5.3.0 (http://surfer.nmr.mgh.harvard.edu/) using the Desikan-Killany atlas parcellation (*Desikan et al., 2006*).

## fMRI preprocessing

Resting state fMRI data were preprocessed with SPM12 using a standard pipeline. First, slice time correction was applied to correct for differences in the time of slice acquisition. EPIs were then realigned to the first EPI acquired, and realignment parameters (translation and rotation) were output. Next, each EPI was coregistered to the native space T1 image. This native space, coregistered fMRI data was used for the EC seed time series extraction (see details below). fMRI data was then warped to MNI space (2 mm isotropic) using the DARTEL templates created during structural preprocessing. Unsmoothed, MNI space fMRI data was used for the alEC and pmEC seed time series extraction (see details below). Spatial smoothing was then performed using a Gaussian kernel of FWHM of 4 × 4×4 mm. Smoothed, MNI space fMRI data was used to conduct the seed-to-voxel FC analyses (see details below).

Next, all resting state fMRI data (including native space, unsmoothed MNI, and smoothed MNI) were optimized for FC analyses using the CONN functional connectivity toolbox (version 17 f) (*Whitfield-Gabrieli and Nieto-Castanon, 2012*) implemented in Matlab version 2015a (The MathWorks, Inc, Natick, MA) for YA and OA samples separately. ART motion detection was first performed to identify volumes of high motion, using a conservative movement threshold of >0.5 mm/TR and a global intensity z-score of 3. Outlier volumes were flagged and included as spike regressors during the denoising process (*Lemieux et al., 2007*; *Power et al., 2015*). We did not exclude any participants from analyses due to excess motion, as all participants had <20% of outlier volumes, with an average of 3.4 ± 2.5% outlier volumes for the YA sample, and an average of 4.9 ± 3.6% outlier volumes for the OA sample. Denoising was then performed with the following parameters included: realignment parameters (translations and rotations) and their first-order derivatives, spike motion regressors, and anatomical CompCor (first five components of time series signal from white matter and CSF) (*Behzadi et al., 2007*). A band pass filter of 0.008–0.1 Hz and linear detrending were then applied to the residual time series.

## Seeds for functional connectivity analysis

### Entorhinal cortex (EC) seed

The EC seed was obtained from the FreeSurfer segmentation of each participant's native space 3T T1 image. This segmentation includes the medial bank of the collateral sulcus, and thus likely contains the transentorhinal region (*Desikan et al., 2006*; *Taylor and Probst, 2008*). Each individual's EC ROIs were masked by native space gray matter masks to ensure the voxels included had a high likelihood of being gray matter. This resulted in no more than 8% of voxels removed in the YA, and

16% of voxels removed in the OA sample. All seeds were manually inspected for quality. Seeds were then resliced using SPM to match the spatial resolution of the native space resting state fMRI data (2.6 mm isotropic). To address signal drop out within the EC seeds, we identified voxels with a mean intensity of <2 SD from the mean intensity across voxels within the seed, and subsequently removed these voxels (*Libby et al., 2012*; *Maass et al., 2015*). This step resulted in no more than 5% of voxels removed across both the YA and OA samples on average (see *Supplementary file 4* for quantification). Time series for the left and right EC seeds were then extracted from the denoised, native space resting state fMRI data.

## Anterolateral (alEC) and posteromedial (pmEC) EC seeds

The alEC and pmEC seeds were functionally defined in a previous study using high-resolution 7T MRI (see *Maass et al., 2015*) for full details). Briefly, in that study the anatomical borders of the whole EC were first manually defined on a high-resolution T1-group template. Multivariate classification was then used to identify clusters of voxels within this mask that showed preferential FC with the perirhinal cortex (alEC ROI) or parahippocampal gyrus (pmEC ROI) in a group of younger adults. These ROIs were warped to MNI template (2 mm) and made publicly available for use in subsequent studies. In the current study, we used these unilateral MNI space ROIs as seeds for FC. Signal drop out within these seeds was addressed with the same method as described for the EC above, but using the MNI space fMRI data to calculate voxel intensity, again resulting in no more than 5% of voxels excluded from the seeds on average (see *Supplementary file 4* for quantification). For a visual representation of the alEC and pmEC seeds on the group-mean functional image, see *Figure 1—figure supplement 2*. Because the alEC and pmEC seeds are spatially adjacent to each other, we extracted time series from the unsmoothed, denoised MNI space resting state data to prevent signal from each seed to be smoothed into the other, which may have obscured results.

## Functional connectivity analysis

### First level analysis

Seed-to-voxel FC analysis was performed with the CONN toolbox using the smoothed MNI space fMRI data. Semi-partial correlations were used for all first-level analyses to determine the unique variance of each seed, controlling for the variance of all other seed regions entered into the same model. We first built a model to determine the FC of left and right EC seeds, controlling for the time series of the contralateral seed, by entering the EC-left and EC-right seeds into the same model. We then built a model to determine the functional connectivity of each EC subregion seed, controlling for the time series of all other EC subregion seeds, by entering the alEC-left, alEC-right, pmEC-left, pmEC-right seeds into the same model. The use of semi-partial correlations in this subregion analysis additionally helped control for the possibility of signal bleed in between alEC and pmEC seeds due to their spatially adjacent locations. All analyses were performed within an explicit mask derived to remove regions of signal dropout across the whole brain. To create this mask, we first calculated a mean functional MNI space image across all participants, and then masked it by a group level gray matter mask. We then removed voxels that had less than 40% of the mean signal intensity of the image. Voxels removed due to signal dropout occurred within inferior lateral temporal, medial temporal, and medial prefrontal regions (see *Figure 1—figure supplement 2* for a visual representation).

### Second-level analysis

To obtain group level patterns of FC, we performed a one sample t-test, controlling for age and sex as covariates of no interest, for the YA and OA samples separately. This resulted in a group level t-map for each unilateral seed, reflecting a positive contrast, with a voxel-level threshold of p<0.001 uncorrected and a cluster level threshold of p<0.05 with FDR correction. We then extracted the within-hemisphere thresholded t-map for each seed, and combined right and left hemispheres for each respective seed to produce the final group level FC t-maps. These final group level FC t-maps were binarized for tau-related analyses. Second-level analyses also resulted in group level beta maps for each seed, where the beta value in each voxel represents the group average Fisher's z' transformed correlation coefficient between the time series of the seed and that voxel. We similarly

extracted within-hemisphere beta maps for each seed, and combined them to produce final group level beta maps for measures of 'FC strength'.

## Functional connectivity post-processing

We removed each seed region from their respective FC mask to ensure that FC results did not reflect autocorrelations, and due to our interest in assessing FC between the EC and non-EC cortical regions. To match the resolution of the fMRI data, we smoothed each seed with a Gaussian kernel of FWHM of 4.77 mm, calculated by accounting for both the original resolution of the data (2.6 mm) and the extra spatial smoothing applied (4 mm). The smoothed seeds were then thresholded to include voxels that contained >15% influence from the seed, which ensured expanded coverage around the original seed region. These expanded seed regions were then removed from their respective FC mask.

To examine associations between FC strength and tau deposition, we classified each seed's FC mask into regions of low, medium, and high FC strength. To do this, we extracted the group-average beta value from each voxel within each final group level FC mask. The beta value at each voxel represents group-average Fisher's z' transformed correlation coefficient between the time series of the seed and that voxel. Because the beta values were non-normally distributed, we performed segmentation of the beta values into FC strength regions in a data driven manner rather than picking arbitrary cut-offs. We applied one-dimensional k-means clustering to the beta values within each FC mask, using the package 'Ckmeans.1d.dp' (*Wang and Song, 2011*), implemented in R version 3.5.1 (http:// www.r-project.org/). Like traditional k-means clustering, one-dimensional k-means clustering classifies data into groups with minimum variability within each group, but across one dimension (*Wang and Song, 2011*). This analysis resulted in each beta value being classified into one of three groups, each representing either low, medium, or high beta values, and therefore FC strength. We then created a separate FC strength mask for the voxels contained within each FC strength group, and calculated the proportion of suprathreshold FTP voxels within each FC strength mask.

## PET and 1.5T structural MRI

### 1.5T structural MRI acquisition

Structural MRIs used for standard PET preprocessing were acquired on a 1.5T Siemens Magnetom Avanto scanner (Siemens, Inc) at Lawrence Berkeley National Laboratory (LBNL). A whole brain high-resolution T1-weighted volumetric magnetization prepared rapid gradient echo image (MPRAGE) was collected (1 mm isotropic voxels, TR = 2110 ms, TE = 3.58 ms, FA = 15).

### PET acquisition

PET was acquired for all OA participants at LBNL. Tau deposition was assessed using [18]F-Flortaucipir (FTP; previously known as [18]F-AV-1451), synthesized at the Biomedical Isotope Facility at LBNL as previously described (*Schöll et al., 2016*). Data was collected on a BIOGRAPH PET/CT Truepoint six scanner (Siemens, Inc) with one of two acquisition schemes: either 75–115 min post-injection or 0–100 and 120–150 min post-injection. Data was subsequently binned into 4 × 5 min frames from 80 to 100 min post-injection CT scans were performed before the start of each emission acquisition. Aβ was assessed using [11]C-Pittsburgh Compound B (PiB), synthesized at the Biomedical Isotope Facility at LBNL (*Mathis et al., 2003*). Data was collected on either the BIOGRAPH scanner or an ECAT EXACT HR scanner (Siemens, Inc). Data was acquired across 35 dynamic frames for 90 min post-injection (4 × 15, 8 × 30, 9 × 60, 2 × 180, 10 × 300, and 2 × 600 s). Either a CT scan or a 10 min transmission scan was performed. All PET images were reconstructed using an ordered subset expectation maximization algorithm, with attenuation correction, scatter correction, and smoothing using a Gaussian kernel of 4 mm.

### 1.5T structural MRI processing

The 1.5T structural images underwent the same processing pipeline as the 3T MRI images used for fMRI, including tissue segmentation, creation of a group specific template with DARTEL, warping to 2 mm MNI space, and creation of an average gray matter mask. Additionally, native space structural MRIs were segmented with FreeSurfer. These ROIs were used to extract FTP in the EC in native space, for partial volume correction of the FTP data, and for calculation of the global PiB index.

## PET processing

FTP images were processed using SPM12. Images were realigned, averaged, and coregistered to the participant's 1.5T structural MRI. Standardized uptake value ratio (SUVR) images were calculated by averaging mean tracer uptake over the 80–100 min data and normalized by an inferior cerebellar gray reference region (*Baker et al., 2017*). Native space SUVR images were warped to MNI space (2 mm) using a DARTEL template produced from the 1.5T structural data. No additional spatial smoothing was applied. Additionally, the mean SUVR of each ROI (structural MRI FreeSurfer segmentation) was extracted from the native space images. This ROI data was partial volume corrected using a modified Geometric Transfer Matrix approach (*Rousset et al., 1998*) as previously described (*Baker et al., 2017*). The mean, partial volume corrected EC FTP SUVR was used in subsequent analyses.

PiB images were processed with SPM12. Images were realigned, averaged across frames from the first 20 min of acquisition, and coregistered to the participant's 1.5T structural MRI. Distribution volume ratio (DVR) images were calculated with Logan graphical analysis over 35–90 min data and normalized by a whole cerebellar gray reference region (*Logan, 2000*; *Price et al., 2005*). Global PiB was calculated across cortical FreeSurfer ROIs as previously described (*Mormino et al., 2012*), and a threshold of 1.065 was used to classify participants into Aβ- and Aβ+ groups. To calculate FC mask specific levels of PiB, native space PiB images were warped to MNI space (2 mm) using a DARTEL template, and the mean PiB DVR within the FC mask was extracted. One participant was missing PiB DVR data, and thus this participant was excluded from any analyses involving measures of PiB or Aβ status.

## FTP Post-processing

To address concerns of off-target binding of the FTP tracer impacting voxelwise analyses (*Marquié et al., 2015*), we removed regions susceptible to off-target binding by creating a 'cleaning mask' that included: (1) removing subcortical regions known to exhibit off-target binding (i.e. caudate, putamen, pallidum, accumbens, thalamus, and cerebellum), (2) removing the choroid plexus, and parts of the hippocampus in close proximity to the choroid plexus, by applying a mask where the ventricles were masked and smoothed, and (3) removing regions of off-target FTP 'hot spots', which were identified by creating a mean FTP image of all Aβ- participants and removing any voxels with a group-mean SUVR of >1.4. This cleaning mask was then smoothed with a Gaussian kernel of FWHM of 4 mm, and thresholded to include voxels that contained >15% influence from the cleaning mask. Voxels within this cleaning mask were then removed from both the FC masks and the voxelwise FTP data.

To match the voxelwise FTP data to the FC masks, we also applied the explicit mask used for the FC analyses to the FTP data. This was applied to ensure that regions removed due to signal dropout, and thus not containing FC, did not contribute to calculations of 'outside cortical regions' FTP levels. Including regions of signal dropout would have biased the 'outside cortical regions' data, because it is possible that FC would have been detected in these regions had there been no signal dropout (e.g. inferior temporal lobe). Finally, to ensure we were analyzing gray matter voxels from the FTP images, we applied a gray matter mask derived from the OA FTP sample to both the FTP images and to the FC masks.

To quantify tau deposition, we used the proportion of suprathreshold FTP voxels (>1.4 SUVR). The proportion of FTP above this value has been previously demonstrated to be the most reliable marker of AD-related tau pathology (*Maass et al., 2017*). An additional benefit of using the proportion of suprathreshold FTP voxels rather than a measure such as mean SUVR is that measurements are not confounded by different region sizes. We quantified the proportion of suprathreshold FTP voxels on an individual participant basis by calculating the number of voxels within a region with FTP SUVR >1.4, and dividing by the total number of voxels in the region. This calculation was performed for each FC mask, cortical regions outside of the EC FC mask, cortical regions outside of a combined alEC/pmEC FC mask, and within FC strength clusters for each seed.

## Statistical analysis

We compared demographic information between the full sample of OA with PET and the subsample of OA who additionally received fMRI. Categorical variables were compared with Chi-squared tests,

and continuous variables were compared with independent samples t-tests. Analyses were performed using SPSS version 25.

Repeated measures ANCOVAs were performed using SPSS. In cases where sphericity was violated, degrees of freedom and corresponding p-values were corrected using Greenhouse-Geisser estimates. Post-hoc analyses further investigating the main effects were performed using paired-samples t-tests, and investigation of Aβ status interactions were performed using independent-samples t-tests. We did not further explore any effects of age or sex, as they were entered into the model as covariates of no interest. ANCOVA main effects and interactions, as well as post-hoc t-tests, were considered significant at a $p < 0.05$ threshold.

We explored associations between EC tau and cortical tau using voxelwise regressions across all OA participants with SPM12. The mean partial volume corrected FTP SUVR was extracted from separate left and right hemisphere EC ROIs for each OA participant. For each unilateral EC ROI, we entered mean FTP SUVR as the predictor, age and sex as covariates of no interest, and the voxelwise FTP images as the dependent variables. Results were thresholded at the voxel level ($p < 0.001$ uncorrected) and at the cluster level (cluster size >100 voxels). To be consistent with the FC results, we then extracted left hemisphere results for the left hemisphere EC ROI, and right hemisphere results for the right hemisphere EC ROI, and combined them to produce the final voxelwise regression t-map.

We then investigated the voxelwise correlation between FC strength and the EC-cortical tau associations. We first took the results of the EC-cortical tau regression analysis, where the beta value at each voxel represented the strength of association between FTP in the EC and in that voxel, and masked it with the significant FC mask of each seed. We next extracted (1) the beta value within each FC mask, representing FC between the seed region and that voxel, and (2) the beta value within the masked EC-cortical tau map, representing the association between EC FTP and that voxel's FTP. For each FC seed, we performed a correlation across all the beta values from the FC mask and the beta values from the masked EC-cortical tau regression with both Pearson's and Spearman's correlation to account for the distribution of the data.

## Acknowledgements

Avid Radiopharmaceuticals enabled the use of the [18]F-Flortaucipir tracer, but did not provide direct funding and were not involved in data analysis or interpretation.

## Additional information

### Funding

| Funder | Grant reference number | Author |
|---|---|---|
| National Institutes of Health | R01-AG034570 | William J Jagust |
| National Institutes of Health | F32-AG057107 | Theresa M Harrison |
| Helmholtz-Gemeinschaft | Postdoc Program PD-306 | Anne Maass |
| Tau Consortium | | William J Jagust |

The funders had no role in study design, data collection and interpretation, or the decision to submit the work for publication.

### Author contributions

Jenna N Adams, Conceptualization, Software, Formal analysis, Methodology, Writing—original draft, Writing—review and editing; Anne Maass, Theresa M Harrison, Suzanne L Baker, Software, Methodology, Writing—review and editing; William J Jagust, Conceptualization, Funding acquisition, Writing—original draft, Writing—review and editing

### Author ORCIDs

Jenna N Adams (iD) https://orcid.org/0000-0002-6702-3851
Theresa M Harrison (iD) https://orcid.org/0000-0002-2036-3496

Suzanne L Baker (iD) https://orcid.org/0000-0003-0209-3127
William J Jagust (iD) http://orcid.org/0000-0002-4458-113X

### Ethics

Human subjects: All participants provided written informed consent. This study was approved by the Lawrence Berkeley National Laboratory institutional review board (protocol # 073H026).

### Decision letter and Author response

Decision letter https://doi.org/10.7554/eLife.49132.025
Author response https://doi.org/10.7554/eLife.49132.026

## Additional files

### Supplementary files

• Source data 1. Source data for the FC specific tau deposition correlations presented in *Supplementary file 2*.
DOI: https://doi.org/10.7554/eLife.49132.018

• Supplementary file 1. Results of ANCOVA models investigating tau deposition within regions of functional connectivity from the older adult sample.
DOI: https://doi.org/10.7554/eLife.49132.019

• Supplementary file 2. Associations between functional connectivity specific tau deposition and both Aβ and EC tau.
DOI: https://doi.org/10.7554/eLife.49132.020

• Supplementary file 3. Results of ANCOVA models comparing tau deposition in different functional connectivity strength regions using older adult FC.
DOI: https://doi.org/10.7554/eLife.49132.021

• Supplementary file 4. Quantification of voxels removed from seed regions due to signal drop out.
DOI: https://doi.org/10.7554/eLife.49132.022

• Transparent reporting form
DOI: https://doi.org/10.7554/eLife.49132.023

### Data availability

Data analyzed during this study are available as supporting files. Source data files have been provided for Table 1, Figure 2, Figure 2-figure supplement 1, Figure 3, Figure 3-figure supplement 1, and Supplementary File 2.

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
