## [Decision Letter]

Thank you for submitting your article "Cortical tau deposition follows patterns of entorhinal functional connectivity in aging" for consideration by *eLife*. Your article has been reviewed by Huda Zoghbi as the Senior Editor, a Reviewing Editor, and two reviewers. The following individuals involved in review of your submission have agreed to reveal their identity: Scott Small (Reviewer #1).

The reviewers have discussed the reviews with one another and the Reviewing Editor has drafted this decision to help you prepare a revised submission.

Summary:

In this elegant and rigorous study, the authors combine state-of-the-art neuroimaging in a large sample of participants to address, and then clarify, a question about tau spread in the brain. They show, with pinpoint precision, how a particular subregion of the entorhinal cortex, one that accumulates alterations in the tau protein first and foremost in Alzheimer's disease (AD), is differentially associated with tau accumulation in cortical brain regions that interconnect with this entorhinal subregion. While this observation has been suggested, or at least could be intuited, by previous postmortem and in vivo neuroimaging studies, this study is distinct in a number of ways, including that: It does so by parsing neighboring entorhinal cortex subdivisions, one more vulnerable to Alzheimer's than the other; and, that remarkably interindividual variability in measures of connectivity are correlated with the degree to which tau is observed in the connected regions. These and other strengths of the study supports their conclusion "that patterns of entorhinal cortex connectivity predict the spatial topography and level of cortical tau deposition in a sample of cognitively normal older adults, where regions with high entorhinal connectivity show more tau." Nevertheless, the manuscript can be improved by addressing the following key points:

Essential revisions:

1) The elegance of the study-nicely illustrating the meaningful insight that can be gleaned from multi-model neuroimaging correlations in human subjects-is dampened by their attempt to overinterpret, particularly regarding mechanisms to explain their findings. Here are few examples:

a) In the Abstract they say, "These results provide an explanation for the anatomic specificity of tau deposition in the aging brain and point to mechanisms of the transition from normal aging to AD." Their results test whether tau spread out of the entorhinal cortex is linked to entorhinal cortex connectivity. It seems to provide very little explanations for why tau begins in the entorhinal cortex in an anatomically specific manner, nor does it really provide a mechanistic explanation for the transition from normal aging to AD. Suggesting that spread is associated with amyloid accumulation is not a true nor novelly-proposed mechanism. And in any case, they show that amyloid is neither necessary nor sufficient for this spread. (More on this below).

b) In the Introduction they say, "The goal of this study was to examine whether and how tau spreads out of the EC through neuronal pathways." In the best case, their correlational studies address 'whether' tau spreads through neuronal pathways, but there really very little novel mechanistic insight of 'how' it does so.

c) In the Discussion section, the authors dedicate paragraphs on mechanism, many that have come from the rodent literature. While a Discussion section can, and should, accommodate some degree of interpretation and even speculation, the Discussion section, as written, seems a bit over-interpretative and over-speculative based on their own elegant findings. It is very hard to infer mechanism without any manipulations, and none exist in this manuscript. The true strength of this manuscript-and it has many-is that by using sophisticated and combined neuroimaging techniques they can map and establish associations in the living human brain. A forced attempt at mechanistic insight is not required for an elegant study of this kind, one that in contrast to artificial mouse studies teaches us what actually happens in the human brain.

2) In fact, they can rely on their elegant human results to test or refute hypotheses that have come from the rodent literature, which while might be more mechanistic in nature, nevertheless suffer from the limitations of mouse models in which multiple genes are typically overexpressed at non-physiological level, and/or in combinations that never exist in humans. For example, they interpret their findings to be supportive of mouse studies, that typically overexpress at non-physiological levels of mutations in tau that occur in FTD not AD, that suggest that tau spreads via anterograde 'axonal projections' to the dentate gyrus. This in fact makes sense in the mouse models, since the dentate gyrus receives the greatest axonal input from the entorhinal cortex. The problem is that, as shown in postmortem human brains, the dentate gyrus is the one hippocampal region that accumulates the least and slowest accumulation of tau pathology. With the wonderful spatial resolution of their sophisticated neuroimaging techniques, the authors should test whether the dentate gyrus, or even just the whole hippocampus, has high tau levels, levels that are dependent on entorhinal cortex connectivity. If not, as suggested by their results, and as can be further confirmed by additional analyses, they can significantly contribute to our understanding by concluding that anterograde axonal is not the determining factor. Perhaps more than anterograde axonal spread, the spread of tau is mediated via retrograde spread, which might better account for their results.

3) Along the same lines, they anchor their mechanistic interpretation upon mouse studies that suggest that tau spread is mediated by increased "neuronal activity". The problem here is that most studies suggest that the entorhinal cortex has dampened not increased neuronal activity in AD (a reliable finding, which I believe is even shown by this group's previous work). Additionally, increased neuronal activity seems to be linked to brain regions where a lot of amyloid accumulates without concomitant accumulation of tau pathology. A number of recent papers have established that tau pathology alone causes a decrease in neuronal activity, and that when co-occurs with amyloid pathology, the dampening effect of tau overcomes the potential stimulatory effect of amyloid. The net effect is decreased neuronal activity. This is particularly relevant to the entorhinal cortex in sporadic AD, where amyloid accumulation in the entorhinal cortex always occur after tau accumulation. Here again, they are encouraged to rely on human findings to inform on the true disease and the limitations of mechanistic rodent studies, and not the other way around.

4) Perhaps the most interesting and novel findings of their study is that amyloid is neither necessary nor sufficient to mediate tau spread out of the entorhinal cortex. This agrees with recent studies where amyloid reducing therapies, which significantly reduce amyloid, nevertheless do not arrest tau spread. In fact, if understood correctly from Figure 3C, they show that for some subregions of the entorhinal cortex amyloid reduces tau spread over its connected regions. This is a fascinating observation. Instead of trying to shoehorn their findings to agree with other papers, the authors are encouraged to feature this novel and important observation to inform the field.

5) The main region of interest the authors use is a notoriously difficult region to properly acquire signals from using fMRI due to susceptibility artifacts. The approach to address this relies on a previous publication which focused on young adults. Although the dissociation between the two putative subnetworks connected to the subregions of the entorhinal cortex can be observed, the present pattern in young adults is less distinct relative to prior reports from other labs. This suggests that perhaps the entorhinal seeds are not optimal for dissociating the relevant connectivity. Since resting-state fMRI connectivity is not directional, an alternative approach would use seed regions distal to regions affected by susceptibility artifacts. This is also critical with older adults showing atrophy in the brain (entorhinal cortex and elsewhere). Improving the dissociation entorhinal-connected regions would confirm more conclusively that tau deposition correlation with resting-state fMRI is not due to this region being less affected by signal loss and could potentially improve the overall results. In addition, the authors should consider presenting visually the coverage in EPI space of these regions and quantification of the quality in both young and older adults.

6) As the authors note, the use of older adults resting-state fMRI poses several challenges which they overcome by using young adults to define and characterize the networks (in parallel to their older adults' results). In Figure 4 and Figure 4—figure supplement 1 the authors report on correlation between deposition measures and group average FC strength per voxel for young and older adults, respectively. One prediction that emerges from the reported results is that variation in tau/Aβ deposition would co-vary with functional connectivity across individuals for network nodes. The absence of such a correlation would suggest that sensitivity is lacking which would be informative in-and-of-itself. Its presence would lend further support to the conclusion that tau deposition tracks functional connectivity and that the latter can be potentially used for tracking tau deposition progression. The authors should carry out this type of analysis.

---

## [Author Response]

Summary:In this elegant and rigorous study, the authors combine state-of-the-art neuroimaging in a large sample of participants to address, and then clarify, a question about tau spread in the brain. They show, with pinpoint precision, how a particular subregion of the entorhinal cortex, one that accumulates alterations in the tau protein first and foremost in Alzheimer's disease (AD), is differentially associated with tau accumulation in cortical brain regions that interconnect with this entorhinal subregion. While this observation has been suggested, or at least could be intuited, by previous postmortem and in vivo neuroimaging studies, this study is distinct in a number of ways, including that: It does so by parsing neighboring entorhinal cortex subdivisions, one more vulnerable to Alzheimer's than the other; and, that remarkably interindividual variability in measures of connectivity are correlated with the degree to which tau is observed in the connected regions. These and other strengths of the study supports their conclusion "that patterns of entorhinal cortex connectivity predict the spatial topography and level of cortical tau deposition in a sample of cognitively normal older adults, where regions with high entorhinal connectivity show more tau." Nevertheless, the manuscript can be improved by addressing the following key points:Essential revisions:1) The elegance of the study-nicely illustrating the meaningful insight that can be gleaned from multi-model neuroimaging correlations in human subjects-is dampened by their attempt to overinterpret, particularly regarding mechanisms to explain their findings. Here are few examples:

We agree with the reviewers’ suggestion that we have focused too much on mechanistic explanations that this study was not designed to answer. We therefore have removed much of the mechanistic discussion from the manuscript and have toned down our interpretations of mechanisms in the Discussion section as suggested. We now focus more on the novel insights that our human neuroimaging results suggest, and elaborate on our findings in more detail.

a) In the Abstract they say, "These results provide an explanation for the anatomic specificity of tau deposition in the aging brain and point to mechanisms of the transition from normal aging to AD." Their results test whether tau spread out of the entorhinal cortex is linked to entorhinal cortex connectivity. It seems to provide very little explanations for why tau begins in the entorhinal cortex in an anatomically specific manner, nor does it really provide a mechanistic explanation for the transition from normal aging to AD. Suggesting that spread is associated with amyloid accumulation is not a true nor novelly-proposed mechanism. And in any case, they show that amyloid is neither necessary nor sufficient for this spread. (More on this below).

We thank the reviewers for pointing out this discrepancy in the Abstract. For the “anatomic specificity of tau deposition…” sentence, we intended to speak to the specificity of *neocortical* tau in the aging brain. As the reviewers pointed out, our study does not provide insight on why tau deposition begins in the entorhinal cortex, however, we believe our findings help explain neocortical vulnerability to tau. The wording of the Abstract has been changed to reflect this (“…anatomic specificity of *neocortical* tau deposition…”).

We agree that our study does not provide a truly mechanistic explanation on how the transition between aging and AD occurs. However, we believe our findings provide some explanation for the observation that tau spread (which is a feature of AD) occurs on the background of normal aging, and this is actually the point we were trying to make. We clarify this in the Abstract, and this point is also mentioned in the Discussion section.

b) In the Introduction they say, "The goal of this study was to examine whether and how tau spreads out of the EC through neuronal pathways." In the best case, their correlational studies address 'whether' tau spreads through neuronal pathways, but there really very little novel mechanistic insight of 'how' it does so.

We again agree with the reviewers on this point and have removed “how” from that sentence in the Introduction.

c) In the Discussion section, the authors dedicate paragraphs on mechanism, many that have come from the rodent literature. While a Discussion section can, and should, accommodate some degree of interpretation and even speculation, the Discussion section, as written, seems a bit over-interpretative and over-speculative based on their own elegant findings. It is very hard to infer mechanism without any manipulations, and none exist in this manuscript. The true strength of this manuscript-and it has many-is that by using sophisticated and combined neuroimaging techniques they can map and establish associations in the living human brain. A forced attempt at mechanistic insight is not required for an elegant study of this kind, one that in contrast to artificial mouse studies teaches us what actually happens in the human brain.

We thank the reviewers for their support of our approach, and have now rewritten the Discussion section to focus more on our results and on the insights gained from our study rather than the previous focus on mechanisms and findings from rodent literature.

2) In fact, they can rely on their elegant human results to test or refute hypotheses that have come from the rodent literature, which while might be more mechanistic in nature, nevertheless suffer from the limitations of mouse models in which multiple genes are typically overexpressed at non-physiological level, and/or in combinations that never exist in humans. For example, they interpret their findings to be supportive of mouse studies, that typically overexpress at non-physiological levels of mutations in tau that occur in FTD not AD, that suggest that tau spreads via anterograde 'axonal projections' to the dentate gyrus. This in fact makes sense in the mouse models, since the dentate gyrus receives the greatest axonal input from the entorhinal cortex. The problem is that, as shown in postmortem human brains, the dentate gyrus is the one hippocampal region that accumulates the least and slowest accumulation of tau pathology. With the wonderful spatial resolution of their sophisticated neuroimaging techniques, the authors should test whether the dentate gyrus, or even just the whole hippocampus, has high tau levels, levels that are dependent on entorhinal cortex connectivity. If not, as suggested by their results, and as can be further confirmed by additional analyses, they can significantly contribute to our understanding by concluding that anterograde axonal is not the determining factor. Perhaps more than anterograde axonal spread, the spread of tau is mediated via retrograde spread, which might better account for their results.

We agree with the reviewers that the question of anterograde versus retrograde tau spread is one of great importance and interest. However, while we agree that contrasting tau accumulation in the dentate gyrus or hippocampus with tau accumulation in neocortex would be desirable, we unfortunately cannot do this for two reasons.

First, while [18F] Flortaucipir (FTP) is a valid tracer for tau aggregation (Marquié et al., 2015), it exhibits known off-target binding in the ventral portion of the choroid plexus (Baker et al., 2017), which spills over into the anterior hippocampus due to the resolution of PET. This phenomenon makes quantifying and interpreting tau deposition within the hippocampus unreliable. Many laboratories, including our own, have slowly shied away from quantifying tau in the hippocampus for this reason. We are concerned that even partial volume correction or approaches using covariance do not solve this problem (Baker et al., 2017; Sperling et al., 2019). In fact, we now mask out regions of the hippocampus that may have received spill over of signal from the choroid plexus in all of our analyses (elaborated on in Materials and methods section). For clarification, we have added this limitation to the Discussion section. The rest of the medial temporal lobe, and in particular the entorhinal cortex, is spared from the effects of this off-target binding and reliable measures of tau deposition can be obtained (Schöll et al., 2016).

Second, the resolution of our resting-state fMRI data (2.6mm isotropic) is not high enough to measure short-range connectivity between the entorhinal cortex and hippocampus. We are currently acquiring higher resolution fMRI data (1.5mm isotropic), and plan to investigate questions of entorhinal subregion to hippocampal subfield connectivity and their relationship to measures of medial temporal lobe tau deposition in a future project.

While the fMRI resolution limits our ability to determine EC-hippocampal connectivity, we still believe this resolution is acceptable to investigate connectivity between alEC/pmEC and other medial temporal and cortical regions. Prior to conducting the study, we had some initial concerns of whether we would be able to differentiate alEC versus pmEC signal with our resolution. Because of this, we also conducted all analyses using the “full” entorhinal seed. Our concerns were alleviated after seeing the distinct patterns of cortical connectivity that replicate previous work. We note that our resolution is lower than previous studies using 7T fMRI, and highlight this limitation and its implications in the Discussion section.

While we currently may not be able to answer the question of anterograde versus retrograde tau spread, we now mention the importance of investigating this question in the Discussion section, and suggest that future neuroimaging studies should aim to resolve this debate.

3) Along the same lines, they anchor their mechanistic interpretation upon mouse studies that suggest that tau spread is mediated by increased "neuronal activity". The problem here is that most studies suggest that the entorhinal cortex has dampened not increased neuronal activity in AD (a reliable finding, which I believe is even shown by this group's previous work). Additionally, increased neuronal activity seems to be linked to brain regions where a lot of amyloid accumulates without concomitant accumulation of tau pathology. A number of recent papers have established that tau pathology alone causes a decrease in neuronal activity, and that when co-occurs with amyloid pathology, the dampening effect of tau overcomes the potential stimulatory effect of amyloid. The net effect is decreased neuronal activity. This is particularly relevant to the entorhinal cortex in sporadic AD, where amyloid accumulation in the entorhinal cortex always occur after tau accumulation. Here again, they are encouraged to rely on human findings to inform on the true disease and the limitations of mechanistic rodent studies, and not the other way around.

We thank the authors for this comment, and are aware of the studies finding reduced neuronal activity in association with tau deposition, regardless of the presence of amyloid. Although we thought relating our connectivity strength results to findings of activity level was an interesting parallel, we realize that this is not an accurate comparison. Furthermore, our connectivity data really do not speak to the question of activity levels. Thus, we have removed this comparison from both the Results section and the Discussion section.

4) Perhaps the most interesting and novel findings of their study is that amyloid is neither necessary nor sufficient to mediate tau spread out of the entorhinal cortex. This agrees with recent studies where amyloid reducing therapies, which significantly reduce amyloid, nevertheless do not arrest tau spread. In fact, if understood correctly from Figure 3C, they show that for some subregions of the entorhinal cortex amyloid reduces tau spread over its connected regions. This is a fascinating observation. Instead of trying to shoehorn their findings to agree with other papers, the authors are encouraged to feature this novel and important observation to inform the field.

We agree with the reviewers that the amyloid-related results are interesting. In light of this, we further elaborate on these concepts in our Discussion section. We now emphasize the finding that amyloid is not necessary to observe relationships between tau and connectivity, and examine our findings in the context of the failure of amyloid reducing therapies.

We also want to further clarify Figure 3C/F. In this figure, although there is less tau deposition in regions of the highest pmEC functional connectivity compared to regions of low or medium functional connectivity, there is still overall more tau deposition within all connected regions in the Aβ+ than the Aβ- participants (Main effect of AB status: F(2)=16.02, p<0.001). In this sense, we are unable to say that “amyloid reduces tau spread over its connected regions”. We have rewritten the description of the pmEC amyloid interaction in the manuscript to help make interpretation of this result more clear.

5) The main region of interest the authors use is a notoriously difficult region to properly acquire signals from using fMRI due to susceptibility artifacts. The approach to address this relies on a previous publication which focused on young adults. Although the dissociation between the two putative subnetworks connected to the subregions of the entorhinal cortex can be observed, the present pattern in young adults is less distinct relative to prior reports from other labs. This suggests that perhaps the entorhinal seeds are not optimal for dissociating the relevant connectivity. Since resting-state fMRI connectivity is not directional, an alternative approach would use seed regions distal to regions affected by susceptibility artifacts. This is also critical with older adults showing atrophy in the brain (entorhinal cortex and elsewhere). Improving the dissociation entorhinal-connected regions would confirm more conclusively that tau deposition correlation with resting-state fMRI is not due to this region being less affected by signal loss and could potentially improve the overall results. In addition, the authors should consider presenting visually the coverage in EPI space of these regions and quantification of the quality in both young and older adults.

Our responses to the specific points within this comment are below:

Signal Drop Out in EC ROIs

We agree with the reviewers about the importance of signal dropout in the entorhinal cortex. We were similarly concerned about this at the start of the project, and thus took many steps to select what we thought was the most appropriate method to address this issue. Initially, we compared two methods to remove voxels of low signal within our entorhinal ROIs, and compared the quality of the resulting functional connectivity maps.

The first method we tested to remove low signal from the entorhinal ROIs was based upon the group-level explicit mask that we applied to all second-level FC and tau-related analyses. To create this explicit mask that removed regions of low signal, we used the group-mean mean functional image and excluded all voxels that had <40% of the mean signal across the gray matter. This resulted in a mask that removed voxels of low signal from regions such as the inferior frontal lobe, inferior temporal lobe, and lateral portions of the medial temporal lobe. We then applied this mask to our entorhinal ROIs, thus removing voxels that had low average signal. This resulted in approximately 10-13% of voxels removed from the alEC, and 0-2% of voxels removed from the pmEC. We created Author response image 1 to allow reviewers to examine a visual representation of the seeds and how they are affected by this approach.

**Author response image 1. respfig1:** Visualization of alEC and pmEC seeds using the group-level explicit mask method to remove signal drop out. The alEC and pmEC seeds are overlaid on the explicit mask (yellow). Regions of low signal (brown) were removed from these seeds to test functional connectivity. The resulting functional connectivity maps are shown in Author response image 2. We also show the seeds overlaid on the group-mean functional image (middle), and the group-mean functional image without the seeds (right).

While this method worked well for the template space alEC and pmEC seeds, we could not apply it to the full EC seed, because that seed was subject-specific and in native space. To determine an approach to address signal dropout in native space, and on a subject-by-subject basis, we applied the method first detailed in Libby et al., 2012 (and subsequently used by Maass et al., 2015). This method removes voxels of low signal on a subject-by-subject basis, which we thought may have advantages over the previous group-level method due to accounting for individual variability in low signal within each ROI. We then applied this method to the EC seed, as well as the alEC and pmEC seeds, to be consistent across our seed regions. Similar to its performance in the previous studies, we found that this method removed no more than approximately 10% of voxels across any of our subjects.

Because we had a choice between the group-level explicit masked ROIs (Method 1) and the subject-level “Libby method” ROIs (Method 2) for the alEC and pmEC ROIs, we compared the resulting functional connectivity maps. As demonstrated in Author response image 2, which we created for reviewers to visually compare how these approaches affect results, the spatial extent and the intensity of our findings looked nearly identical. This suggested that both methods worked similarly in addressing signal drop out. Thus, to be consistent across our ROIs, we chose to apply the Libby et al., 2012 method to not only the EC seed, but to the alEC and pmEC seeds. We believe using a consistent method makes the results from the full EC seed and from the alEC/pmEC subregions more comparable. As suggested, we now report quantification (mean and range) of the proportion of voxels retained using this approach as a supplemental table (see Supplementary file 4). We have also created a supplemental figure in which we overlay the alEC and pmEC seeds on the group-mean functional image for a visual representation of the average signal (see Figure 1—figure supplement 2). This is similar to Author response image 1 but does not show the seeds overlaid on the explicit mask because we did not use that approach in the manuscript, and that visualization was created for the reviewers to compare methods.

**Author response image 2. respfig2:** Comparison of signal drop out methods for the alEC and pmEC seeds, and their resulting functional connectivity maps. Functional connectivity maps for Method 1 (group-level explicit mask, see Author response image 1) and Method 2 (subject-level from Libby et al., 2012) of removing signal drop out from alEC/pmEC seeds were nearly identical. Quantification of voxels retained is shown below each image. Due to the similarity of the methods, and wanting consistency with the EC seed, we used Method 2 in the manuscript.

We also applied the group-level explicit mask described above to all of our functional connectivity analyses. This ensured that no regions within our group-level connectivity maps exhibited low signal, and that no correlations were due to similarly low signal in the seeds and in the target voxels. At the helpful suggestion of the reviewers, we now show a visualization of the explicit mask and the extent of voxels removed due to low signal for our functional connectivity analyses in Figure 1—figure supplement 2.

Overall, we are confident that our findings are not an artifact due to signal drop out. If anything, signal loss is greater in the lateral aspects of the entorhinal cortex compared to the medial (see Author response image 1 and/or Figure 1—figure supplement 2). Thus, we believe that the lack of findings between pmEC FC and tau are not due to weak signal, but to a true null effect.

Subnetwork Distinction

We respectfully disagree with the reviewers’ comment that the alEC/pmEC subnetworks are not distinct. The only other study that has investigated cortical connectivity of alEC and pmEC (Navarro Schröder et al., 2015) showed patterns of connectivity that were highly similar to our young adult results. In addition, work from the Ranganath Lab and the Buckner Lab (Kahn et al., 2008; Libby et al., 2012) showed network dissociations using perirhinal and parahippocampal cortical seeds that were highly similar to the networks we observed. While we previously presented our subnetworks visually and found minimal regions of overlap within the healthy young adult sample (Figure 1F), we have now conducted an analysis to test this overlap statistically. We compared the subnetwork maps using the Dice Similarity Coefficient (DSC), and found a value of 0.11 for the young adult FC maps, which indicates poor spatial overlap (Savio et al., 2017). While we believe this minimal overlap does not prevent our functional connectivity patterns from being described as “distinct”, we now elaborate more within the limitations section of the Discussion section about how future studies at higher resolution would help to further investigate the specificity of these subnetwork patterns.

In the older adults, the overlap between the subnetworks is slightly increased (DSC=0.17), though still minimal, and reflects the visual increase of overlap (Figure 1—figure supplement 1). The cortical connectivity of alEC and pmEC has not previously been described for older adults, and thus the overlap we observe may be a true change rather than indicating a problem with our seed regions. As stated in the Discussion section, we believe this increased overlap in the OA may either be due to dedifferentiation in aging, or to methodological concerns about the fit of these template-space seeds in older adult subjects that may have atrophy. We discuss both possibilities, and due to our uncertainty of the underlying cause, we do not focus on the older adult data and only present it as supplementary data. We plan to further investigate older adult FC of the entorhinal cortex more precisely in future studies, using native space alEC/pmEC seeds that are segmented from high resolution T2 data that is currently being collected.

Use of Distal Seeds

While using distal seed regions to overcome limitations of signal drop out is an interesting suggestion, we believe it is not appropriate for the current study for a number of reasons. First, the use of the perirhinal cortex and parahippocampal gyrus in place of the alEC and pmEC, respectively, (as conducted in Libby et al., 2012, Kahn et al., 2008), would not resolve the concern of signal dropout. Signal dropout is strongest in the lateral portion of the medial temporal lobe, and thus the perirhinal cortex would suffer from similar levels of signal dropout as the alEC. Furthermore, we chose alEC and pmEC because these regions directly reflect the location of the earliest tau deposition in the medial temporal lobe. Therefore, substituting the alEC seed with the perirhinal cortex would not definitively solve this concern.

Second, the use of even more distal seeds outside of the MTL would alter the resulting patterns of connectivity and not be representative of the true connectivity from the alEC and pmEC subregions. While it is true that functional connectivity is non-directional and thus reciprocal between two structures, each structure has its own unique network of connectivity. Even though canonical intrinsic networks like the default mode network can be generated from multiple different seed regions, the alEC and pmEC are unique in that they are the apex of bidirectional processing streams and thus ideally require the seeds to be placed locally to enable characterization of their networks. For example, while pmEC may be functionally and reciprocally connected to the retrosplenial cortex (RSC), using RSC as the seed in a seed-to-voxel analysis would also reveal widespread connectivity with structures that are uniquely connected to RSC and do not exhibit connectivity with pmEC.

Finally, and most importantly, we believe using distal seeds to the alEC and pmEC would not follow the motivation behind our study, which is to investigate whether these very specific patterns of FC derived from the earliest regions of tau deposition are associated with cortical tau spread.

6) As the authors note, the use of older adults resting-state fMRI poses several challenges which they overcome by using young adults to define and characterize the networks (in parallel to their older adults' results). In Figure 4 and Figure 4—figure supplement 1 the authors report on correlation between deposition measures and group average FC strength per voxel for young and older adults, respectively. One prediction that emerges from the reported results is that variation in tau/Aβ deposition would co-vary with functional connectivity across individuals for network nodes. The absence of such a correlation would suggest that sensitivity is lacking which would be informative in-and-of-itself. Its presence would lend further support to the conclusion that tau deposition tracks functional connectivity and that the latter can be potentially used for tracking tau deposition progression. The authors should carry out this type of analysis.

We agree that investigating the relationship between functional connectivity and tau deposition at the individual level is an intriguing question. However, we were somewhat uncertain as to the exact analysis requested. It is not possible, for example, to examine subject-level functional connectivity measures because these are too noisy at the individual level. However, to respond to this concern we did examine how these group entorhinal functional connectivity strength patterns predicted individual level patterns of tau deposition (see Author response image 3).

To determine whether group-level functional connectivity strength patterns were able to predict subject-level patterns of tau deposition, we performed a voxelwise correlation for each individual older adult subject. We correlated the FC strength (β value) at each voxel (either YA or OA FC; analogous to Figure 4A-C y-axis) with each individual subject’s FTP SUVR at the same voxel. This produced a correlation coefficient for each individual OA subject representing how well patterns of FC strength with entorhinal seeds predicted their own pattern of tau deposition (Author response image 3, each data point = one subject’s correlation coefficient). Using the YA FC, these predictions were significantly greater than zero for the EC and alEC, indicating that voxels with stronger connectivity were associated with higher levels of tau at the single-subject level (Author response image 3; one sample t-test, EC: t(122) = 14.45, p<0.001; alEC: t(122) = 16.98, p<0.001), while pmEC had an inverse relationship (pmEC: t(122) = -6.11, p<0.001). These patterns generally repeated when using the OA FC, though the alEC was no longer significantly greater than zero (Author response image 3AB; EC: t(122) = 11.26, p<0.001; alEC: t(122) = 1.27, p = 0.21; pmEC: t(122) = -22.78, p<0.001).

**Author response image 3. respfig3:** Functional connectivity strength predicting individual-level patterns of tau deposition. For the young adult (**a**) and older adult (**b**) connectivity, we ran voxelwise correlations between group-average connectivity strength (β) and individual voxelwise FTP SUVR. This resulted in a correlation coefficient per subject indicating how well patterns of FC predicted their individual pattern of tau (each data point = one subject’s correlation). Distributions of the correlations were compared to zero with one-sample t-tests. c-h, To explore the variability in these correlation results, and to parallel findings from Figure 4 of the main manuscript, we correlated each subjects correlation strength between FC and FTP (data points from a and b) with their mean EC FTP. We found positive correlations, indicating that as a subject’s mean EC tau increased, patterns of FC better predicted their individual pattern of cortical tau deposition. * p<0.05 **p<0.01 ***p<0.001.

Next, to try to explain the variability of this association across subjects, and to parallel the findings in Figure 4 of the main manuscript, we correlated each subject’s individual correlation strength (between FC strength and their individual pattern of tau deposition, from Author response image 3) with the mean amount of EC FTP SUVR in each subject, controlling for age and sex (see Author response image 3). We found a positive association across subjects, such that subjects with more EC tau had their neocortical tau better predicted by patterns of FC. This suggests that as tau in the EC increases, it spreads out of the EC in a pattern consistent with its FC at the single subject level.

We believe these findings are consistent with the group-level findings in Figure 4, but on an individual-level tau basis. While these individual-level results are interesting, we do not think these findings fundamentally change the conclusions of the current study, and thus we have presented the results here to the reviewers rather than adding them to the main manuscript. However, if the editors and/or reviewers would like us to add these individual-level results to the main manuscript or as supplementary material, we are happy to do so, and will revise the manuscript further to include the analyses presented here.